# Consistency Flow Model Achieves One-step Denoising Error Correction Codes

## Abstract

Error Correction Codes (ECC) are fundamental to reliable digital communication, yet designing neural decoders that are both accurate and computationally efficient remains challenging. Recent denoising diffusion decoders with transformer backbones achieve state-of-the-art performance, but their iterative sampling limits practicality in low-latency settings. We introduce the *Error Correction Consistency Flow Model (ECCFM)*, an architecture-agnostic training framework for high-fidelity one-step decoding. By casting the reverse denoising process as a Probability Flow Ordinary Differential Equation (PF-ODE) and enforcing smoothness through a differential time regularization, ECCFM learns to map noisy signals along the decoding trajectory directly to the original codeword in a single inference step. Across multiple decoding benchmarks, ECCFM attains lower bit-error rates (BER) than autoregressive and diffusion-based baselines, with notable improvements on longer codes, while delivering inference speeds up from 30x to 100x faster than denoising diffusion decoders.

## 1 Introduction

Error Correction Codes (ECC) play a central role in modern digital communications and have been essential in a wide range of applications, including wireless communication and data storage. The core task of an ECC decoder is to recover a message from a received signal corrupted by noise during transmission. Recently, inspired by the great success of deep neural networks in domains such as computer vision (He et al., 2016), generative modeling (Goodfellow et al., 2020; Ho et al., 2020; Song et al., 2020b), and natural language processing (Vaswani et al., 2017; Brown et al., 2020), neural network-based ECC decoders were introduced, which have been shown to be capable of improving the performance scores of conventional, problem-specific algorithms such as Belief Propagation (BP)(Richardson & Urbanke, 2002) and Min-Sum (MS)(Fossorier et al., 1999).

The existing neural decoders can be categorized into two groups. Early model-based approaches (Lugosch & Gross, 2017; Nachmani & Wolf, 2019; Zhu et al., 2020; Dai et al., 2021; Kwak et al., 2022) achieved successful results by integrating neural networks into conventional decoding algorithms. However, their reliance on problem-specific structures can limit their applicability as a general-purpose decoder. To address this limitation and train a general-purpose decoder, model-free decoders enable the extension of a general neural bet architecture without prior knowledge of the decoding algorithm. While the early proposals applying fully connected neural net architectures (Gruber et al., 2017; Cammerer et al., 2017; Kim et al., 2018) could lead to overfitting, the preprocessing techniques that utilize magnitude and syndrome vectors (Bennatan et al., 2018) have been effective in reducing the error. These improvements have led to efficient transformer-based decoders (Choukroun & Wolf, 2022b), which leverage the self-attention mechanism to improve the numerical performance on short block codes. The auto-regressive method in transformer-based decoders has been improved in several recent works (Choukroun & Wolf, 2024a;b). Notably, the recent work by Park et al. (2025) integrates cross-attention between magnitude and syndrome, resulting in the state-of-the-art performance in the decoding task.

In parallel, the remarkable success of diffusion generative models (Ho et al., 2020; Song et al., 2020b) across various domains (Rombach et al., 2022; He et al., 2022; Lou et al., 2023; Chen et al., 2024; Nie et al., 2025) has inspired a new direction for deep learning-based ECC. Diffusion models train a noise estimator to gradually denoise a noisy input data and reverse a forward noise

manipulation process, generating high-quality samples by an iterative application of a neural net denoiser to an input Gaussian noise. The DDECC method (Choukroun & Wolf, 2022a) extends the denoising diffusion framework to ECC, naturally modeling the AWGN channel as a forward diffusion process. In this setup, a time-dependent transformer learns to denoise the received signal iteratively, recovering the original codeword. This framework provides complementary gains over auto-regressive methods and establishes a new state-of-the-art in terms of Bit Error Rates (BER), particularly for long codes and low SNR, due to its iterative refinement denoising process.

However, the iterative denoising of DDECC has introduced a new challenge: the multi-step sampling process incurs significant computational overhead and high latency, reducing the practicality of the approach in real-world applications. This motivates the following question: Can a decoder maintain the performance achieved by the denoising diffusion frameworks at a lower latency that meets the latency requirements of several communication settings?

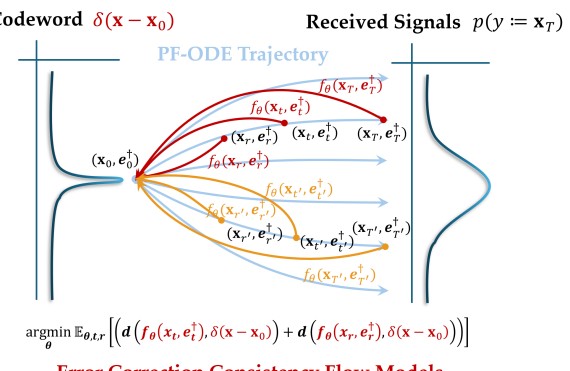

Figure 1: Illustration of our proposed Error Correction Consistency Flow Model (ECCFM). ECCFM learns to map received signals from the trajectories to a single, consistent codeword prediction $\mathbf{x}_0$, represented by a $\delta$ function.

In this work, we aim to address this question by proposing Error Correction Consistency Flow Models (ECCFM), an architecture-agnostic training framework designed to construct high-fidelity, one-step denosing decoders. Inspired by recent advances in consistency models (Song et al., 2023), we formulate the decoding process as a Probability-Flow Ordinary Differential Equation (PF-ODE), which seeks to learn an optimal trajectory from the noisy signal distribution to the clean codeword distribution as shown in Figure 1. However, a key challenge in the ECC setting is that the noise level cannot be measured by a continuous timestep as in image generation; instead, it is indicated by the sum of syndrome error in (Choukroun & Wolf, 2023). A direct adaptation of consistency models struggles to learn this highly non-smooth decoding trajectory.

To overcome the mentioned challenge, we propose a differential time condition, using a soft-syndrome formulation from (Lau et al., 2025) to regularize the reverse ODE process. This ensures the decoding trajectory is smooth for single-step mapping. Building upon this theoretical foundation, the ECCFM decoder is trained directly with a consistency objective and a soft-syndrome regularization term, enabling it to map any noisy received signal to the estimated original codeword in a single, efficient inference step.

We demonstrate the effectiveness of ECCFM through several numerical experiments on a diverse set of standard codes, including BCH, Low-Density Parity-Check (LDPC), Polar, MacKay, and CCSDS. The results show that ECCFM can improve Bit-Error-Rate (BER) compared to leading auto-regressive methods, exhibiting particularly strong gains on longer codes with code lengths above 200. Notably, it achieves an inference speedup of 30x to 100x over the existing iterative denoising diffusion methods in a single step, while maintaining comparable decoding performance. In our numerical evaluation, ECCFM offers a model-free training diagram for building ECC decoders that achieves near state-of-the-art performance with low latency, and requires only a single-step inference suited for most real-world applications.

## 2 RELATED WORKS

**Neural Network-based ECC Decoders.** Neural network-based decoders are broadly categorized into model-based and model-free approaches. Model-based decoders augment conventional algorithms, such as Belief Propagation (BP)(Richardson & Urbanke, 2002) and Min-Sum (MS)(Fossorier et al., 1999), by using neural networks to learn the message-passing process. This paradigm has been extensively explored across various architectures and code types (Dai et al.,

2021; Kwak et al., 2022; 2023; Lugosch & Gross, 2017; Nachmani & Wolf, 2019; 2021; Marinkovic et al., 2010; Zhu et al., 2020), consistently achieving superior performance over conventional algorithms (Matsumine & Ochiai, 2024). However, they are often limited by challenges in capturing long-range dependencies and the reliance on the underlying decoding algorithm.

In contrast, model-free decoders treat decoding as a learning problem without depending on problem-specific algorithms. While early fully-connected architectures (Gruber et al., 2017; Cammerer et al., 2017; Kim et al., 2018) struggled with overfitting, (Bennatan et al., 2018) proposes the pre-processing by decomposing the magnitude and syndrome vector to address overfitting issues for model-free decoders. Inspired by the recent breakthrough of Transformers (Vaswani et al., 2017), ECCT (Choukroun & Wolf, 2022b) pioneered by applying self-attention to the channel output, and modeling decoding as an auto-regressive sequence-to-sequence task. Several works have improved based on this approach: FECCT (Choukroun & Wolf, 2024a) improved generalization, DC-ECCT (Choukroun & Wolf, 2024b) enabled joint encoder-decoder training, and CrossMPT (Park et al., 2025) introduced cross-attention to achieve better performance and efficiency among auto-regressive methods. Recently, diffusion generative models have emerged as a powerful alternative. DDECC (Choukroun & Wolf, 2023) frames decoding as a denoising process, modeling the AWGN channel as the forward diffusion step, and offers performance gains over auto-regressive decoders.

**Diffusion Generative Models.** Diffusion generative models (Sohl-Dickstein et al., 2015; Ho et al., 2020; Song et al., 2020b; Karras et al., 2022) learn to reverse a forward noising process by estimating the data's score function (Stein, 1972). They have achieved state-of-the-art performance in synthesizing high-fidelity samples across diverse domains like images (Dhariwal & Nichol, 2021; Rombach et al., 2022; Podell et al., 2024), video (He et al., 2022; Blattmann et al., 2023), text generation (Lou et al., 2023; Nie et al., 2025), and graphs (Sun & Yang, 2023; Li et al., 2023; Lei et al., 2025). However, a primary limitation of diffusion models is the significant computational overhead during inference, due to the iterative nature of the denoising process. To mitigate this, numerous accelerated sampling methods have been developed (Song et al., 2020a; Lu et al., 2022).

This framework is particularly well-suited for Error Correction Code (ECC) decoding, where the AWGN channel naturally models the forward process (Choukroun & Wolf, 2022a). While diffusion-based decoders have reached state-of-the-art performance (Choukroun & Wolf, 2022a; Park et al., 2024), they inherit the same computational inefficiency compared to methods like auto-regressive decoders (Choukroun & Wolf, 2022b; Park et al., 2024). To address this issue, we draw inspiration from Consistency Models (CMs) (Song et al., 2023; Song & Dhariwal, 2023; Geng et al., 2024), a novel technique for accelerating diffusion models. CMs are designed to directly learn the reverse denoising trajectory, enabling mapping noisy samples to the target data distribution in one step and maintaining high generation quality.

## 3 PRELIMINARIES

**Error Correction Codes.** In the error correction code setting, we consider a linear codebook $\mathcal{C}$, defined by a $k \times n$ generator matrix $G$ and an $(n-k) \times n$ parity-check matrix $H$. Note that these matrices satisfy $GH^\top = 0$ over the binary field $\mathbb{F}_2$. The encoder maps a message $m \in \{0,1\}^k$ to an $n$-bit codeword $x \in \mathcal{C} \subset \{0,1\}^n$ via the linear transformation $x = mG$. The codeword $x$ is then modulated using Binary Phase-Shift Keying (BPSK), where $0 \mapsto +1$ and $1 \mapsto -1$, resulting in the signal $x_s \in \{-1,+1\}^n$. We suppose this signal is transmitted over an Additive White Gaussian Noise (AWGN) channel. The received signal $y$ is given by: $y = x_s + z$, where the noise vector $z$ is sampled from an isotropic Gaussian distribution, $z \sim \mathcal{N}(0, \sigma^2 I_n)$.

The objective of a decoder is to estimate the original codeword $\hat{x}$ from the noisy signal $y$. An essential tool for error detection is the syndrome, calculated from a hard-demodulated formulation of the received signal, $y_b$, performed as $y_b = \text{bin}(\text{sign}(y))$. Here $\text{sign}(y)$ is $+1$ for $y \geq 0$ and $-1$ otherwise, and the bin function maps $\{-1,+1\}$ back to $\{1,0\}$. The syndrome is then computed as $s = H y_b^\top$. An error is detected if the syndrome is a non-zero vector (i.e., $s(y) \neq 0$). Following the pre-processing technique proposed by (Bennatan et al., 2018), the input vector to the neural network is $[|y|, s(y)]$ with length $n + (n-k)$ to avoid overfitting.

**Diffusion Generative Models**. We provide a detailed review of diffusion generative models in the Appendix, Section A.1.

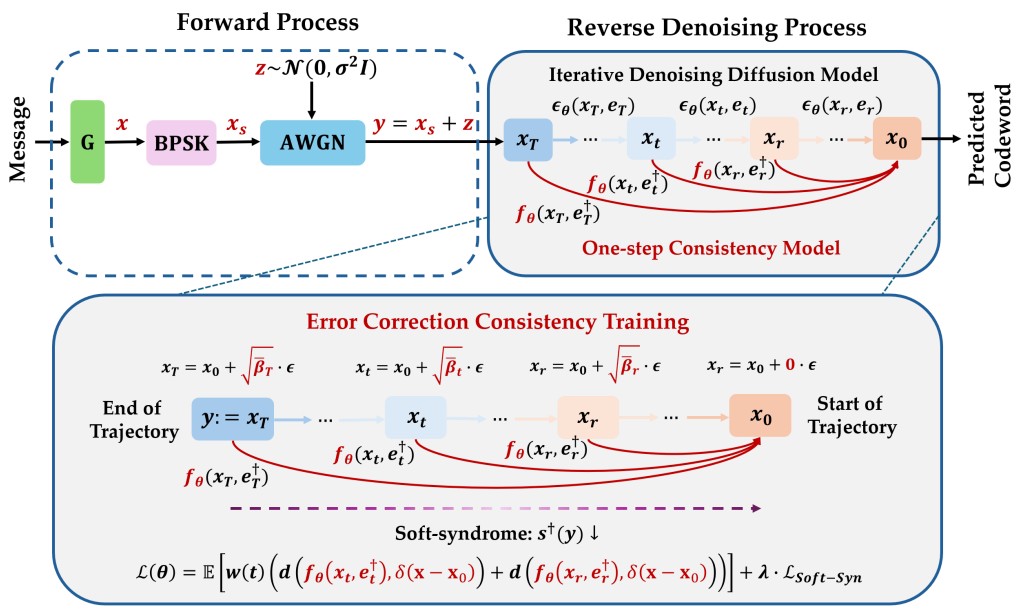

Figure 2: Training Dynamics from iterative denoising to 1-step consistency decoding. DDECC's iterative diffusion denoising learns a noise predictor, $\epsilon_\theta(\cdot, e_t)$, requiring a multi-step iterative process to reverse the noise and decode the codeword. Our ECCFM, $f_\theta(\cdot, e^\dagger)$, directly learns the mapping from any noisy signal to the original clean codeword. By using the smooth soft-syndrome condition $(e^\dagger)$, it achieves successful decoding in a single step.

## 4 METHOD: CONSISTENCY FLOW MODEL FOR ONE-STEP DECODING

### 4.1 ERROR CORRECTION CONSISTENCY FLOW PROPERTY

Despite its impressive decoding performance, a drawback of the DDECC algorithm is the computational overhead during inference due to its iterative denoising mechanism. Inspired by the recent success of Consistency Models (CMs) (Song et al., 2023; Song & Dhariwal, 2023) in image generation, we propose the first approach for training consistency models for Error Correction Codes (ECC). Consistency Models (CMs) (Song et al., 2023) were introduced to overcome the inference computational costs by enabling fast, one-step generation. The core principle is that: any two points $(\mathbf{x}_t, t)$ and $(\mathbf{x}_r, r)$ on the same PF-ODE trajectory should map to the same origin point $\mathbf{x}_0$. CMs build upon Eq. 13 and learn a function $f_\theta(\mathbf{x}_t, t)$ that directly estimates the *trajectory* from noisy data to clean data with a single step:

$$f_\theta(\mathbf{x}_t, t) = \mathbf{x}_0, \tag{1}$$

The training objective of CMs is to enforce the self-consistency property across a discrete time steps. The continuous time interval $[0, T]$ is discretized into $N - 1$ sub-intervals, defined by timesteps $1 = t_1 < \cdots < t_N = T$. The model is then trained to minimize the following loss, which enforces that the model's output remains consistent at adjacent points on the same PF-ODE trajectory:

$$\mathcal{L}_{\text{Standard-CM}}(\theta) := \mathbb{E}[w(t)d\left(f_\theta(\mathbf{x}_t, t), f_\theta(\mathbf{x}_r, r)\right)], \tag{2}$$

Here, $f_\theta$ is the consistency network being trained and $w(t)$ denotes the time schedule. This function is defined by two fundamental properties proposed by (Song et al., 2023; Song & Dhariwal, 2023): **1) Boundary Condition**: At timestep $t = 0$, the function is the identity: $f_\theta(\mathbf{x}_0, 0) = \mathbf{x}_0$. **2) Self-Consistency**: For any two points $(\mathbf{x}_t, t)$ and $(\mathbf{x}_r, r)$ on the same trajectory, the function yields the same output: $f_\theta(\mathbf{x}_t, t) = f_\theta(\mathbf{x}_r, r)$.

Unlike generation tasks, which aim to sample from a target data distribution $p_{\text{data}}$, the goal in ECC is to decode a single, ground-truth codeword $\mathbf{x}_0$ corresponding to a given received noisy signal $y$. The ideal target distribution for the decoder is therefore a point mass at the correct codeword, which we model conceptually as a Dirac delta function, $\delta(\mathbf{x} - \mathbf{x}_0)$. Based on this, we propose that a consistency-based decoder should satisfy the following Error Correction Consistency Flow condition:

**Error Correction Self-consistency.** Given a trajectory $\{y := \mathbf{x}_t\}_{t \in [0,T]}$, we learn the consistency function as $f : (\mathbf{x}_t, t; \theta) \mapsto \mathbf{x}_0$, holding the Error Correction Consistency Flow property: for a given ground-truth codeword $\mathbf{x}_0$, all points $(\mathbf{x}_t, t)$ on any trajectory originating from $\mathbf{x}_0$ map directly back to it, i.e., $f_\theta(\mathbf{x}_t, t) = \mathbf{x}_0, \quad \forall t \in [0, T]$. This implies that for any two noisy signals $\mathbf{x}_t$ and $\mathbf{x}_r$ derived from the same $\mathbf{x}_0$, their consistency function outputs must be identical and correct: $f_\theta(\mathbf{x}_t, t) = f_\theta(\mathbf{x}_r, r) = \mathbf{x}_0, \forall t, r \in [0, T]$. We learn the consistency model such that the conditional distribution is modeled as $p_\theta(\mathbf{x}|\mathbf{x}_t) = \delta(\mathbf{x} - f_\theta(\mathbf{x}_t, t))$, where $f_\theta(\mathbf{x}_t, t)$ directly estimates $\mathbf{x}_0$.

To train a consistency model $f_\theta$ that achieves this property, a naive adaptation of the vanilla consistency loss would be to minimize the distance between two noisy signals from the same trajectory similar to Eq. 2. However, this standard consistency objective is indirect; it only enforces relative consistency between outputs. In decoding tasks, the ground truth $\mathbf{x}_0$ is known during training, and consistency models are optimized to decode the clean codeword. We leverage this property by proposing the Error Correction Consistency Flow Loss (EC-CM), which minimizes the distance of each estimation to the ground truth $\mathbf{x}_0$:

$$\mathcal{L}_{\text{EC-CM}}(\theta) := \mathbb{E}[w(t) \left[ d\left( f_\theta(\mathbf{x}_t, t), \mathbf{x}_0 \right) + d\left( f_\theta(\mathbf{x}_r, r), \mathbf{x}_0 \right) \right]], \tag{3}$$

However, in ECC domain, Binary Cross Entropy (BCE) is the commonly used divergence measure. By Proposition 1, we demonstrate that our formulation in Eq. 3 serves as an upper bound on the vanilla consistency loss in terms of Total variation distance, directly optimizing for the Error Correction Consistency property.

**Proposition 1** *Let $\mathcal{L}_{\text{Standard-CM}}$ be defined by the Total Variation distance, $TV(\cdot, \cdot)$, and $\mathcal{L}_{\text{EC-CM}}$ be defined by the Binary Cross Entropy, $BCE(\cdot, \cdot)$. For any timesteps $t, r$ and ground truth $\mathbf{x}_0$, the following semi–triangle inequality holds:*

$$TV^2\left( f_\theta(\mathbf{x}_t, t), f_\theta(\mathbf{x}_r, r) \right) \leq BCE\left( f_\theta(\mathbf{x}_t, t), \mathbf{x}_0 \right) + BCE\left( f_\theta(\mathbf{x}_r, r), \mathbf{x}_0 \right). \tag{4}$$

This fundamental change requires all decoding trajectories mapping to their origin codeword, as demonstrated in Figure 1, and the learned solution distribution is expected to center on $\mathbf{x}_0$.

## 4.2 DIFFERENTIAL CONSISTENCY CONDITION FOR SMOOTH TRAJECTORY

The training objective of a consistency model is to learn a function $f_\theta(\mathbf{x}_t, t)$ that maps any point on a trajectory back to its origin $\mathbf{x}_0$. This is enforced by both satisfying the self-consistency property and the boundary condition, which is rooted in the differential equation $\frac{df}{dt} = 0$. Following Eq. 10 and 11 in (Geng et al., 2024), the consistency function $f_\theta$ is parameterized to satisfy these conditions:

$$f_\theta(\mathbf{x}_t, t) = \mathbf{x}_0 \Leftrightarrow \frac{df}{dt} = 0, f_\theta(\mathbf{x}_0, 0) = \mathbf{x}_0, \tag{5}$$

In practice, this differential form is discretized for training using a *finite-difference approximation*, by dividing the time horizon into $N - 1$ sub-intervals $1 = t_1 < \cdots < t_N = T$:

$$0 = \frac{df}{dt} \approx \frac{f_\theta(\mathbf{x}_t, t) - f_\theta(\mathbf{x}_r, r)}{t - r}, \tag{6}$$

where $dt = \Delta t = t - r, t > r \geq 0$.

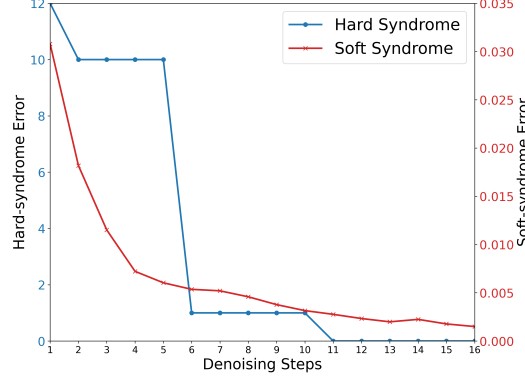

Figure 3: Decoding trajectories for models conditioned on Hard Syndrome versus Soft Syndrome on a POLAR(64,48) code. The soft-syndrome conditioning results in a smoother path to a valid codeword. Additional Results under low SNR are available in Appendix A.5.2, Figure 7.

A critical problem arises when applying the consistency framework to ECC decoding tasks: the time variables $t$ and $r$, which represent noise levels, are not directly observable from the received signals. A seemingly natural solution, proposed in DDECC (Choukroun & Wolf, 2023), is to use the sum of syndrome error, $e_t$, as a measurement of the noise level, since an error count of zero ($e_t = 0$) indicates a valid codeword. The hard syndrome is computed as $s = H y_b^\top$, where $H$ is the parity-check matrix, and the sum of syndrome error, $e_t = \sum_i s(\mathbf{x}_t)_i$, is then the sum of binary syndrome bits.

---

**Algorithm 1** Error Correction Consistency Training

---

**Require:** Model $f_{\boldsymbol{\theta}}$, parity-check matrix $\mathbf{H}$, learning rate $\eta$, syndrome weight $\lambda$, denoising steps $N$, time scaling factor $\alpha$.

   **for** training batch $\mathbf{x}_0$ **do**
      $t \sim \mathcal{U}\{1, \ldots, N\}, r = \alpha t$                    ▷ Sample forward diffusion steps
      $\boldsymbol{\epsilon} \sim \mathcal{N}(\mathbf{0}, \mathbf{I})$                             ▷ Sample Gaussian noise
      $\mathbf{x}_t \leftarrow \mathbf{x}_0 + \sqrt{\bar{\beta}_t} \cdot \boldsymbol{\epsilon}, \mathbf{x}_r \leftarrow \mathbf{x}_0 + \sqrt{\bar{\beta}_r} \cdot \boldsymbol{\epsilon}$      ▷ Generate noisy signals via forward process
      $e_t^{\dagger} = \mathcal{L}_{\text{Soft-syn}}(\mathbf{x}_t, H), \; e_r^{\dagger} = \mathcal{L}_{\text{Soft-syn}}(\mathbf{x}_r, H)$        ▷ Calculate soft-syndrome
      $\mathcal{L}_{\text{EC-CM}} \leftarrow \mathcal{L}_{\text{BCE}}(f_{\theta}(\mathbf{x}_t, e_t^{\dagger}), \mathbf{x}_0) + \mathcal{L}_{\text{BCE}}(f_{\theta}(\mathbf{x}_r, e_r^{\dagger}), \mathbf{x}_0)$
      $\mathcal{L}_{\text{Total}} \leftarrow \mathcal{L}_{\text{Consistency}} + \lambda \cdot \left( \mathcal{L}_{\text{Soft-syn}}(f_{\theta}(\mathbf{x}_t, e_t^{\dagger}), H) + \mathcal{L}_{\text{Soft-syn}}(f_{\theta}(\mathbf{x}_r, e_r^{\dagger}), H) \right)$
      $\boldsymbol{\theta} \leftarrow \boldsymbol{\theta} - \eta \cdot \nabla_{\boldsymbol{\theta}} \mathcal{L}_{\text{Total}}$
   **end for**

---

However, naively replacing the continuous time variables $t$ and $r$ in Eq. 6 with their discrete, integer-valued counterparts $e_t$ and $e_r$ violates the core assumption of a smooth, differential trajectory. As we demonstrate in Figure 3, the trajectory defined by the syndrome error is highly non-smooth and clustered: A small change in the noisy signal can cause an abrupt jump in the syndrome error count, invalidating the finite-difference approximation; And it is common for two different noisy signals, $\mathbf{x}_t$ and $\mathbf{x}_r$, to have the same syndrome error count ($e_t = e_r$), leading to instability during training.

Therefore, the discrete and non-smooth nature of the syndrome error makes it an unsuitable conditioning variable for standard consistency model training in ECC, which necessitates a smooth and differential measure of noise level along the denoising trajectory. We then propose replacing this discrete error sum condition with a continuous and differentiable alternative. Inspired by (Lau et al., 2025), we introduce the *soft syndrome* as the basis for the consistency noise condition. The soft syndrome, $s^{\dagger}$, is a fully differentiable function that leverages the log-likelihood ratios of the received signal $\mathbf{x}_t$: and offers a continuous measure of how close each parity-check equation is to being fulfilled.

**Differential Soft-syndrome Error Time Condition.** Similar to the conventional hard sum of syndrome error, we compute the soft-syndrome error condition $e_t^{\dagger}$ of the parity-check matrix $H$ as:

$$e_t^{\dagger} := -\frac{1}{n-k} \sum_j \log \Pr(S_j^{\dagger} = 0) = -\frac{1}{n-k} \sum_j \log(1 - s_j^{\dagger}), \tag{7}$$

$$\mathcal{L}_{\text{Soft-syn}}(\mathbf{x}_t, H) = -e_t^{\dagger}, \tag{8}$$

This soft-syndrome error condition is computed as the binary cross-entropy between the estimated syndrome and the all-zero syndrome, which requires valid codewords to satisfy all parity-check equations. We use the mean-field approximation to provide differential conditions to estimate the probability of satisfying zero-syndrome conditions in Eq. 7:

$$s_j^{\dagger} = \frac{1}{2} - \frac{1}{2} \prod_{\{i : H_{j,i}=1\}} \left( 2 \cdot \text{sigmoid}\left(\frac{2\mathbf{x}_{t,i}}{\sigma^2}\right) - 1 \right), \quad \Pr(S_j^{\dagger} = 0) = 1 - s_j^{\dagger}, \tag{9}$$

where $x_{t,i}$ is the $i$-th position of the received codeword $x_t$, the noise level $\sigma$ is given to the decoder at the receiver's side. The soft-syndrome is zero if and only if the codeword is valid, yet it varies smoothly and continuously with the received signal $\mathbf{x}_t$. By using $e_t^{\dagger}$ as the time condition, we provide the consistency model with a smooth, differentiable trajectory from a noisy signal to a valid codeword, resolving the instability and degeneracy issues of the hard error sum and enabling stable training.

## 4.3 ERROR CORRECTION CONSISTENCY FLOW TRAINING DYNAMICS

Building upon the differential time conditions via soft-syndrome in Eq. 7, we make it able to learn a smooth trajectory satisfying consistency conditions as shown in Figure 2. Given a codeword

$x_s \in \{-1, +1\}^n$ modulated using BPSK, and the signal received is then perturbed with an AWGN channel $y = x_s + z = x_s \cdot \tilde{z}_s$, where $\tilde{z}_s$ denotes the multiplicative noise. We follow the standard pre-processing techniques proposed by (Bennatan et al., 2018), the input of the neural network is a concatenated vector representing magnitude and hard syndrome $[|y|, s(y)]$ with length $2n - k$. For the forward process in AWGN channel, we build the trajectory by adding the same Gaussian noise $\epsilon \sim \mathcal{N}(0, I)$ with a different time schedule $\sqrt{\bar{\beta}_t}$, where $t \in [0, ..., N]$ and $N$ denotes the pre-defined forward noising steps. Thus, during training, we sample different noisy signals $y_1 := \mathbf{x}_t, y_2 := \mathbf{x}_r$ with different noise levels $t \sim \mathcal{U}\{0, ..., N\}$ and $r = \alpha t$, i.e. $\mathbf{x}_t := \mathbf{x}_0 + \sqrt{\bar{\beta}_t} \cdot \epsilon$, $\mathbf{x}_r := \mathbf{x}_0 + \sqrt{\bar{\beta}_r} \cdot \epsilon$, where $\alpha \in [0, 1]$ denotes the time scaling factor. Then a consistency model $f_\theta$ predicts the clean codeword. Following Eq. 3, we get the consistency loss for two different noisy signals $y_t, y_r$, which learns the mapping to their original clean codeword $\mathbf{x}_0$. We further add the soft-syndrome loss in Eq. 7 as the regularization term to stabilize training validated in Appendix A.5.4 and propose the total loss for ECCFM:

$$\mathcal{L}_{\text{Total}}(\theta) = \mathbb{E}\Big[ w(t) \underbrace{\Big( d(f_\theta(\mathbf{x}_t, e_t^\dagger), \mathbf{x}_0) + d\left(f_\theta(\mathbf{x}_r, e_r^\dagger), \mathbf{x}_0\right) \Big)}_{\text{Consistency Loss}}$$

$$+ \lambda \cdot \underbrace{\Big( \mathcal{L}_{\text{Soft-syn}}(f_\theta(\mathbf{x}_t, e_t^\dagger), H) + \mathcal{L}_{\text{Soft-syn}}(f_\theta(\mathbf{x}_r, e_r^\dagger), H) \Big)}_{\text{Soft-syndrome Loss}} \Big] \tag{10}$$

where $d(\cdot, \cdot)$ is a distance metric, such as Binary Cross-Entropy (BCE) in this work, and $\lambda$ is a hyperparameter that weights the syndrome regularization term. Once trained according to Algorithm 1, the learned consistency function $f_\theta$ can decode the noisy received signal $y$ in a one step: $\hat{\mathbf{x}}_0 = f_\theta(y, e_t^\dagger)$, as shown in Appendix 2, Algorithm 2.

## 5 NUMERICAL RESULTS

**Datasets.** We evaluate our proposed ECCFM framework on the following set of standard error correction codes, including BCH, Polar, and LDPC codes (MacKay, CCSDS, and WRAN variants). Our evaluation considers multiple code lengths ($n$), rates ($k/n$), and Signal-to-Noise Ratios (SNRs), specifically $E_b/N_0$ values from 4 to 6 dB, to ensure a robust assessment of performance.

**Evaluation Metrics.** We evaluate decoding performance using two standard metrics following established benchmarks (Choukroun & Wolf, 2022b; 2023; Park et al., 2025): Bit Error Rate (BER) and Frame Error Rate (FER). BER measures the fraction of individual bits that are incorrectly decoded. FER (also known as Block Error Rate, BLER) measures the fraction of entire codewords that contain one or more bit errors. Concerning the latency factor, we evaluate computational efficiency by reporting inference time and throughput (decoded samples per second).

**Baselines.** We numerically compared the results with multiple baselines for the decoding task, including: 1) Conventional BP-based decoders: BP (Bennatan et al., 2018) and ARBP (Nachmani & Wolf, 2021). 2) Auto-regressive model-free decoders: ECCT (Choukroun & Wolf, 2022b), and CrossMPT (Park et al., 2025). 3) Denoising diffusion model-free decoders: DDECC (Choukroun & Wolf, 2023).

**Experimental Setup.** We reproduced the results for all model-free baselines (ECCT, CrossMPT, DDECC) by implementing them with their publication-stated hyperparameters. Our primary EC-CFM model utilizes a Cross-attention Transformer following CrossMPT, using $N = 6$ layers and a hidden dimension of $d = 128$. All the other three baselines (ECCT, CrossMPT, DDECC) apply the same network architecture to ensure fair comparison. ECCFM was trained for 1500 epochs using the Adam optimizer on a single GPU. The learning rate was managed by a cosine decay scheduler, starting at $10^{-4}$ and decreasing to $5 \times 10^{-7}$. Detailed training configurations and hyperparameter selections are provided in Appendix A.4.

**Overall Performance**. Following established benchmarks (Choukroun & Wolf, 2022b; 2023; Park et al., 2025), we conducted a decoding performance comparison measured in $-\ln(\text{BER})$. Our method was evaluated versus two classes of decoders: conventional BP-based algorithms (BP and ARBP) and model-free neural decoders (ECCT, CrossMPT, and DDECC). To ensure a fair comparison, all neural models were implemented with a fixed architecture ($N = 6$ layers, $d = 128$ hidden

Table 1: Performance comparison of various decoders across different codes and Signal-to-Noise Ratios ($E_b/N_0$). The results are reported in terms of $-\ln(\text{BER})$ (the higher, the better). All model-free methods use a fixed model architecture ($N = 6$, $d = 128$). Best results are shown in **bold** and the second-best results are shown in **underline**, respectively.

| Architecture | | BP-based decoders | | | | | | Model-free decoders | | | | | | | | | | | |
|---|---|---|---|---|---|---|---|---|---|---|---|---|---|---|---|---|---|---|---|
| | | BP | | | ARBP | | | ECCT | | | CrossMPT | | | DDECC | | | ECCFM(Ours) | | |
| Code Type | Parameters | 4 | 5 | 6 | 4 | 5 | 6 | 4 | 5 | 6 | 4 | 5 | 6 | 4 | 5 | 6 | 4 | 5 | 6 |
| BCH | (63,36) | 4.03 | 5.42 | 7.26 | 4.57 | 6.39 | 8.92 | 4.69 | 6.48 | 9.06 | 4.94 | 6.74 | 9.28 | **5.02** | 6.82 | **9.88** | 5.00 | **6.89** | 9.76 |
| | (63,45) | 4.36 | 5.55 | 7.26 | 4.97 | 6.90 | 9.41 | 5.47 | 7.56 | 10.51 | **5.73** | 7.98 | 10.80 | 5.68 | **8.08** | **11.22** | 5.70 | 8.03 | 11.04 |
| POLAR | (64,32) | 4.26 | 5.38 | 6.50 | 5.57 | 7.43 | 9.82 | 6.87 | 9.21 | 12.15 | 7.42 | 9.94 | 13.28 | 7.04 | 9.44 | 12.70 | **7.55** | **10.31** | **13.80** |
| | (64,48) | 4.74 | 5.94 | 7.42 | 5.41 | 7.19 | 9.30 | 6.21 | 8.31 | 10.85 | 6.36 | 8.53 | 11.09 | 5.93 | 8.00 | 10.44 | **6.56** | **8.78** | **11.52** |
| | (128,64) | 4.10 | 5.11 | 6.15 | 4.84 | 6.78 | 9.30 | 5.79 | 8.45 | 11.10 | 7.45 | 9.71 | 14.31 | 7.71 | 11.40 | 13.85 | **8.01** | **12.22** | **16.71** |
| | (128,86) | 4.49 | 5.65 | 6.97 | 5.39 | 7.37 | 10.13 | 6.29 | 8.98 | 12.82 | 7.43 | 10.80 | 15.13 | 7.61 | 10.50 | 13.88 | **7.78** | **11.21** | **16.05** |
| | (128,96) | 4.61 | 5.79 | 7.08 | 5.27 | 7.44 | 10.20 | 6.30 | 9.04 | 12.40 | 7.06 | 10.25 | 13.23 | 7.14 | 10.31 | 13.66 | **7.21** | **10.52** | **14.32** |
| LDPC | (121,60) | 4.82 | 7.21 | 10.87 | 5.22 | 8.31 | 13.07 | 5.12 | 8.21 | 12.80 | 5.75 | 9.42 | 15.21 | 5.42 | 9.11 | 13.82 | **6.02** | **9.94** | **15.55** |
| | (121,70) | 5.88 | 8.76 | 13.04 | 6.45 | 10.01 | 14.77 | 6.30 | 10.11 | 15.50 | 7.06 | 11.29 | 17.10 | 6.91 | 11.02 | 17.15 | **7.35** | **12.23** | **17.60** |
| | (121,80) | 6.66 | 9.82 | 13.98 | 7.22 | 11.03 | 15.90 | 7.27 | 11.21 | 17.02 | 7.87 | 12.65 | 17.72 | 7.61 | 11.89 | 16.18 | **8.25** | **13.33** | **18.69** |
| MacKay | (96,48) | 6.84 | 9.40 | 12.57 | 7.43 | 10.65 | 14.65 | 7.37 | 10.55 | 14.72 | 7.85 | 11.72 | 15.49 | **8.03** | **12.44** | 15.79 | 7.92 | 12.25 | **16.08** |
| CCSDS | (128,64) | 6.55 | 9.65 | 13.78 | 7.25 | 10.99 | 16.36 | 6.82 | 10.60 | 15.87 | 7.56 | 11.87 | 16.80 | 7.77 | 12.35 | **17.22** | **7.95** | **12.68** | 17.01 |

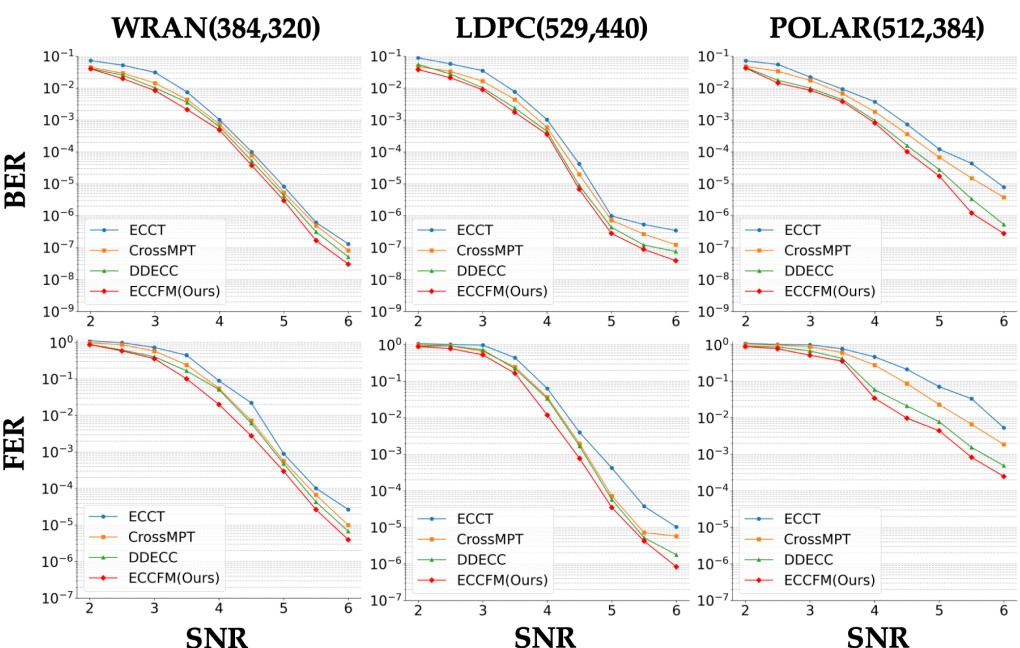

Figure 4: Performance comparison of various decoding baselines on medium-to-long block codes. The plot shows the Bit Error Rate (BER) at different Signal-to-Noise Ratios (SNRs), from 2 dB to 6 dB and divided by 0.5 dB.

dimensions). Furthermore, to ensure statistical significance, each simulation was run until at least 500 error codes were observed, under a maximum of $10^8$ test instances.

As shown in Table 1, our proposed ECCFM framework consistently achieves better performance across the tested code families, including BCH, Polar, LDPC, CCSDS and MacKay with different code rates $(n, k)$. In the test cases, ECCFM achieves the best or second-best BER, performing better than autoregressive and diffusion-based neural decoders and showing considerable gain over POLAR codes. This result indicates the applicability and effectiveness of the proposed consistency-based training approach.

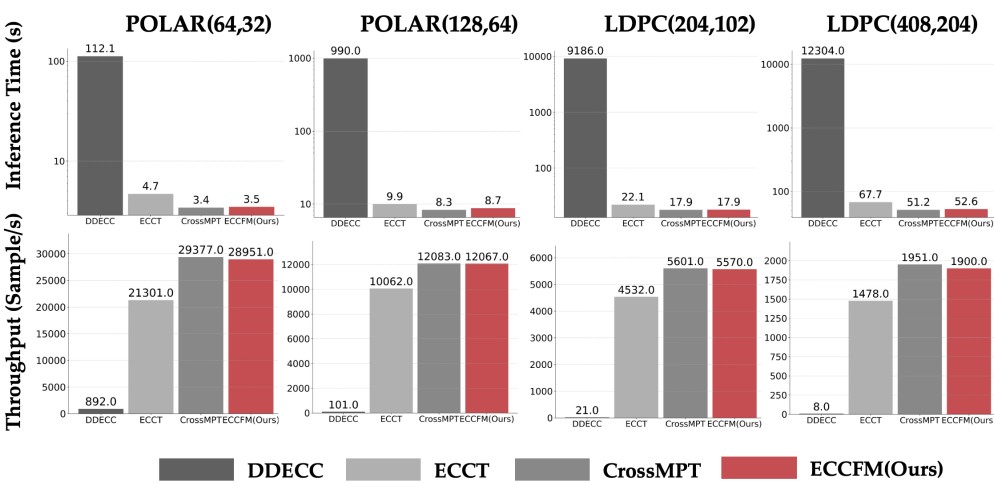

Figure 5: Comparison of Inference Time (top) and Throughput (bottom) across various decoding baselines and code types.

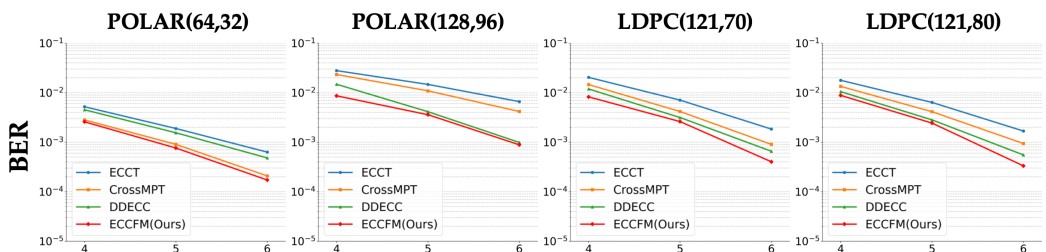

Figure 6: Performance comparison of various decoding baselines on Rayleigh Fading channels. The plot shows the Bit Error Rate (BER) at different Signal-to-Noise Ratios (SNRs).

**Performance on Longer Codes.** To further evaluate scalability, we conducted a focused evaluation on longer codes commonly used in practical communication systems: LDPC($n = 204, k = 102$), LDPC($n = 529, k = 440$), WRAN($n = 384, k = 320$), and Polar($n = 512, k = 384$). All methods were implemented with the same model architecture ($N = 6, d = 128$). As illustrated in Figure 4, ECCFM improves upon the scores of the neural net baselines across a range of SNRs (2 dB to 6 dB), highlighting the scalability and robustness of ECCFM for those challenging decoding tasks. Additional results on other high-length codes are presented in Appendix A.5.7, which also suggest inference speed gain compared to the baselines.

**Performance on Rayleigh Fading Channel.** While our proposed ECCFM training dynamics and the definition of soft-syndrome are intrinsic to the code structure and independent of the underlying channel model, we validate our approach on Rayleigh fading channels to demonstrate robustness. We follow the same experimental settings described in (Choukroun & Wolf, 2022b; Park et al., 2024). In contrast to the AWGN channel, the received signal is defined by: $y = hx + z$, where $h$ is an $n$-dimensional i.i.d. Rayleigh distributed vector with a scale parameter $\alpha = 1$ and $z \sim N(0, \sigma^2)$. In Figure 6, our method achieved competitive performance compared to baselines across different SNRs on Polar and LDPC. These numerical results confirm that our proposed ECCFM framework generalizes effectively to non-Gaussian channels without fundamental modifications.

**Inference Time and Throughput Comparison.** A key benefit of ECCFM is its ability to perform high-fidelity decoding in only one step. To quantify this efficiency gain, we measured inference time (total seconds to decode $10^5$ samples) and throughput (samples decoded per second). As shown in Figure 5, ECCFM demonstrates a speed advantage over diffusion-based methods such as DDECC, achieving speedups of over **30x** for short codes and **100x** for medium-to-long codes. This disparity arises because diffusion models require several iterative denoising steps for inference, a computational cost that scales with code complexity as detailed in Appendix A.5.8. Notably, ECCFM

Table 2: Performance comparison of ECCFM versus the standard ECCT and DDECC, using an identical ECCT backbone on POLAR and LDPC codes.

| Architecture | | ECCT Backbone | | | | | | | | |
|---|---|---|---|---|---|---|---|---|---|---|
| | | ECCT | | | DDECC | | | ECCFM(ECCT) | | |
| Code Type | Parameters | 4 | 5 | 6 | 4 | 5 | 6 | 4 | 5 | 6 |
| POLAR | (64,32) | 6.87 | 9.21 | 12.15 | 7.04 | 9.44 | 12.70 | **7.12** | **9.77** | **12.71** |
| | (64,48) | 6.21 | 8.31 | 10.85 | 5.93 | 8.00 | 10.44 | **6.38** | **8.55** | **11.23** |
| | (128,64) | 5.79 | 8.45 | 11.10 | **7.71** | **11.40** | 13.85 | 7.32 | 11.03 | **14.87** |
| | (128,86) | 6.29 | 8.98 | 12.82 | **7.61** | **10.50** | 13.88 | 7.18 | 10.17 | **15.02** |
| | (128,96) | 6.30 | 9.04 | 12.40 | **7.14** | **10.31** | 13.66 | 6.86 | 9.94 | **13.83** |
| LDPC | (121,60) | 5.12 | 8.21 | 12.80 | 5.42 | **9.11** | 13.82 | **5.55** | 8.86 | **13.97** |
| | (121,70) | 6.30 | 10.11 | 15.50 | **6.91** | 11.02 | **17.15** | 6.87 | **11.21** | 16.13 |
| | (121,80) | 7.27 | 11.21 | 17.02 | 7.61 | 11.89 | 16.18 | **7.80** | **12.03** | **17.95** |

achieves decoding speeds comparable to the fastest auto-regressive baseline (CrossMPT) due to its one-step nature. Therefore, ECCFM matches the competitive performance of denoising diffusion decoders while operating at the high throughput of single-step auto-regressive decoders.

**Ablation Study: Model-Agnostic Property of ECCFM.** We discussed that ECCFM is a model-agnostic training framework, i.e., its performance could be preserved over different neural network architectures. To test this, we conducted an ablation study where we decoupled our framework from the cross-attention transformer backbone conducted before. Specifically, we took the underlying architecture of the ECCT baseline and trained it using our proposed ECCFM training objective. We then compared this model directly against the original ECCT, which uses the same architecture. The results in Table 2 show that applying the ECCFM training objective yields improvement in $-\ln(\text{BER})$ over the standard ECCT, with comparable performance versus the iterative denoising DDECC method. These results demonstrate that ECCFM's property of model-agnostic and its benefits are attributable to the consistency-based training framework itself.

**Ablation Study: Necessity of Soft-syndrome Time Condition in Consistency Training** We further investigated the necessity of using soft-syndrome as a time condition for building the smooth trajectory. We trained ECCFM with either a hard-syndrome time condition or a soft-syndrome time condition. In Appendix A.5.5, Table 7, our experiments revealed that using soft-syndrome as time condition is necessary for successful training of ECCFM. The continuous nature of the soft-syndrome provides a smooth reverse trajectory to learn, whereas models conditioned on the hard syndrome failed to converge.

## 6 CONCLUSIONS AND LIMITATIONS

In this work, we introduced the Error Correction Consistency Flow Model (ECCFM), a novel training framework for obtaining the successful results of diffusion-based decoders with the low latency required for several practical applications. By reformulating the decoding task as a one-step consistency mapping and introducing a differential soft-syndrome error condition, we managed to handle non-smooth trajectories that previously hindered the application of consistency models to ECC. Our experiments demonstrate that ECCFM achieves comparable Bit-Error-Rates across various standard codes, while offering a consistent inference speedup over denoising diffusion methods. Despite these successful results, our work has limitations to be addressed in future research. First, our evaluation is focused on the AWGN channel; the framework's performance and the suitability of the soft-syndrome condition on other channel models, such as fading channels, remain to be investigated. Second, the convergence rate and training efficiency of consistency models highly rely on building a smooth trajectory, which varies in different code types. Future work can explore adaptive methods for more general decoding tasks.

## REPRODUCIBILITY STATEMENT

We ensure that the results presented in this paper are reproducible. The core algorithm has been made available in the supplementary material to facilitate verification. The experimental setup, including training configurations and model architectures, is described in detail in the Appendix. All code types appearing in experiments, such as BCH, POLAR, and LDPC, are publicly available.

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

# A APPENDIX

## A.1 PRELIMINARY ON DIFFUSION GENERATIVE MODELS

**Diffusion Models.** Diffusion Models (DMs) (Ho et al., 2020; Song & Ermon, 2019; Song et al., 2020b) are generative models that generate samples from a target data distribution, $p_{\text{data}}(x_0)$ by reversing a predefined forward noising process (Sohl-Dickstein et al., 2015). In the forward diffusion process, a data sample $x_0$ is gradually perturbed with Gaussian noise over a continuous time interval $t \in [0, T]$. This forward process can be mathematically described as adding noise to obtain a noisy data point $\mathbf{x}_t = \sqrt{\alpha_t}\mathbf{x}_0 + \sqrt{1 - \alpha_t}\boldsymbol{\epsilon}_t$, where $\boldsymbol{\epsilon}_t \sim \mathcal{N}(\mathbf{0}, I)$ is standard Gaussian noise and $\alpha_t \in [0, 1]$ monotonically decreases with time step $t$ to control the noise level. Denoising Diffusion Probabilistic Models (DDPMs) (Ho et al., 2020) $\boldsymbol{\epsilon}_\theta : \mathcal{X} \times [T] \mapsto \mathcal{X}$ is trained to predict the noise $\boldsymbol{\epsilon}_t$ at each time step $t$, also learn the *score function* of $p_t(\mathbf{x_t})$ (Song & Ermon, 2019; Song et al., 2020b):

$$\min_\theta \mathbb{E}_{\mathbf{x}_t, \boldsymbol{\epsilon}_t, t} \left[ \|\boldsymbol{\epsilon}_\theta(\mathbf{x}_t, t) - \boldsymbol{\epsilon}_t\|_2^2 \right] = \min_\theta \mathbb{E}_{\mathbf{x}_t, \boldsymbol{\epsilon}_t, t} \left[ \left\| \boldsymbol{\epsilon}_\theta(\mathbf{x}_t, t) + \sqrt{1 - \alpha_t} \underbrace{\nabla_{\mathbf{x}_t} \log p_t(\mathbf{x}_t)}_{\text{Score Function}} \right\|_2^2 \right], \quad (11)$$

During inference, samples can be generated by solving the reverse-time SDE starting from $t = T$ to $t = 0$. Crucially, there exists a corresponding deterministic process, the *probability flow ODE* (PF-ODE), whose trajectories share the same marginal distributions $p_t(x_t)_{t \in [0,T]}$ as the SDE (Song et al., 2020b). The formulation of PF-ODE can be described and simplified as following (Karras et al., 2022; Song et al., 2023):

$$d\mathbf{x}_t = -\dot{\sigma}(t)\sigma(t)\nabla_{\mathbf{x}_t} \log p_t(\mathbf{x}_t)dt, \quad (12)$$

where $\epsilon_\theta(\mathbf{x}_t, t)$ is the learned time-dependent neural network $\epsilon_\theta(\mathbf{x}_t, t)$, known as the denoiser. , is trained to approximate this expectation: $\epsilon_\theta(x_t, t) \approx \mathbb{E}[x_0 | x_t]$. By substituting this approximation and adopting the common noise schedule $\sigma(t) = t$ following (Karras et al., 2022; Song et al., 2023), the PF-ODE simplifies to:

$$\frac{d\mathbf{x}_t}{dt} = -t\nabla\mathbf{x}_t \log p_t(\mathbf{x}_t) = \frac{\mathbf{x}_t - \epsilon_\theta(\mathbf{x}_t, t)}{t}, \quad (13)$$

Sample generation of diffusion models is performed by solving this PF-ODE backwards in time from $t = T$ to $t = 0$, starting from a sample drawn from the prior Gaussian distribution, $\mathbf{x}_T \sim \mathcal{N}(0, \sigma(T)^2 I)$. This requires a numerical ODE solver (e.g., Euler (Song & Ermon, 2019; Song et al., 2020b) or Heun (Karras et al., 2022)) to obtain a *solution trajectory* $\{\hat{\mathbf{x}}_t\}_{t \in [0,T]}$ that transforms noise into a data sample.

**Denoising Diffusion Error Correction Codes.** The application of diffusion models to error correction was pioneered by DDECC (Choukroun & Wolf, 2023). Its core insight is to model the transmission of a BPSK-modulated codeword $\mathbf{x}_0 \in \{-1, 1\}^n$ over an AWGN channel as the forward diffusion process. A received signal $y$ is treated as a noisy sample $\mathbf{x}_t$ at a specific timestep $t$, where the noise schedule is designed to match the channel's characteristics. This forward process is described as:

$$y := \mathbf{x}_t = \mathbf{x}_0 + \sqrt{\bar{\beta}_t}\epsilon, \quad (14)$$

where $\epsilon \sim \mathcal{N}(0, I)$, and the cumulative noise variance $\bar{\beta}_t = \sum_{i=1}^t \beta_i$ corresponds to the channel's noise level $\sigma^2$. Decoding is then performed via an iterative reverse denoising process, starting with the received signal $y := \mathbf{x}_t$ and applying the denoising update rule for multiple steps, with a trained denoising network $\epsilon_\theta(\cdot, \cdot)$ predicting the multiplicative noise:

$$\mathbf{x}_{t-1} = \mathbf{x}_t - \frac{\sqrt{\bar{\beta}_t}\beta_t}{\bar{\beta}_t + \beta_t}(\mathbf{x}_t - \text{sign}(\mathbf{x}_t)\epsilon_\theta(\mathbf{x}_t, t)), \quad (15)$$

A key innovation in DDECC is its conditioning mechanism. In the ECC domain, the number of parity check errors (syndrome sum), $e_t = \sum_i s(y)_i$, serves as a direct measure of the noisy level in a received signal $y$. Therefore, DDECC conditions its denoising network $\epsilon_\theta$ on the sum of syndrome

error $e_t$ instead of a timestep $t$, making the diffusion models adapted to the structure of the error correction problem. The denoising network is trained to learn the hard prediction of the multiplicative noise with a Binary Cross-Entropy (BCE) loss:

$$\mathcal{L}(\theta) = -\mathbb{E}_{e_t, \mathbf{x}_0, \epsilon} \log(\epsilon_\theta(\mathbf{x}_0 + \sqrt{\bar{\beta}_t}\epsilon, e_t), \tilde{\epsilon}_b), \tag{16}$$

where $\tilde{\epsilon}_b = \text{bin}(\mathbf{x}_0(\mathbf{x}_0 + \sqrt{\bar{\beta}_t}\epsilon))$ denotes the target binary multiplicative noise.

**Consistency Models (CMs).** A major concern of DMs is the slow sampling process, which requires sequential calculation of the denoiser $\epsilon_\theta$. Consistency Models (CMs) (Song et al., 2023) were introduced to overcome this by enabling fast, 1-step generation. The core principle is the self-consistency property: any two points $(\mathbf{x}_t, t)$ and $(\mathbf{x}_r, r)$ on the same PF-ODE trajectory should map to the same origin point, $\mathbf{x}_0$. CMs build upon Eq. 13 and learn a function $f_\theta(\mathbf{x}_t, t)$ that directly estimates the *trajectory* from noisy data to clean data with a single step:

$$f_\theta(\mathbf{x}_t, t) = \mathbf{x}_0, \tag{17}$$

The training objective of CMs is to enforce the consistency property across a discrete set of time steps. The continuous time interval $[0, T]$ is discretized into $N-1$ sub-intervals, defined by timesteps $1 = t_1 < \cdots < t_N = T$. The model is then trained to minimize the following loss, which enforces that the model's output is consistent for adjacent points on the same ODE trajectory:

$$\arg\min_\theta \mathbb{E}[w(t_i)d(f_\theta(\mathbf{x}_{t_{i+1}}, t_{i+1}), f_{\theta-}(\tilde{\mathbf{x}}_{t_i}, t_i))], \tag{18}$$

Here, $f_\theta$ is the network being trained, while $f_{\theta-}$ is an exponential moving average (EMA) of $f_\theta$'s past samples. The term $\tilde{\mathbf{x}}_{t_i} = \mathbf{x}_{t_{i+1}} - (t_i - t_{i+1})t_{i+1}\nabla_{\mathbf{x}_{t_{i+1}}} \log p_{t_{i+1}}(\mathbf{x}_{t_{i+1}})$ is obtained by taking a single ODE solver step backwards from $\mathbf{x}_{t_{i+1}}$ using the score function. This training process can be performed in two ways: by distilling knowledge from a pre-trained diffusion model, known as Consistency Distillation (CD), or by training from scratch, known as Consistency Training (CT). However, training CMs is difficult and resource-intensive. It requires a carefully designed curriculum for the number of discretization steps $N$ to ensure stabilized training. The follow-up works improved the vanilla CMs, such as iCT (Song & Dhariwal, 2023), which proposed enhanced metrics and schedulers, and ECT (Geng et al., 2024), which uses "pre-training diffusion + consistency tuning" to stabilize learning.

### A.2 CONSISTENCY SAMPLING ALGORITHM

Once we obtain the well-trained consistency model $f_\theta$ following Algorithm 1, we can simply apply the one-step sampling to estimate the codeword $\hat{\mathbf{x}}_0$, given any received signals $y$.

---

**Algorithm 2** Error Correction Consistency One-step Sampling

---

**Require:** Consistency Model $f_\theta$, parity-check matrix $\mathbf{H}$.
  **for** Test batch noisy signals $y$ **do**
    $e_t^\dagger = \mathcal{L}_{\text{Soft-syn}}(y, H)$                         ▷ Calculate soft-syndrome
    $\hat{\mathbf{x}}_0 = f_\theta(y, e_t^\dagger)$                   ▷ Estimate clean codeword with one-step
  **end for**

---

### A.3 PROOF OF PROPOSITION 1

**Proof 1** *Let $\hat{\mathbf{x}}_t = f_\theta(\mathbf{x}_t, t)$ and $\hat{\mathbf{x}}_r = f_\theta(\mathbf{x}_r, r)$ denote the predicted outcome by consistency model at timesteps $t$ and $r$, respectively. Let $\mathcal{L}_{\text{Standard-CM}}$ be defined by the Total Variation distance, $TV(\cdot, \cdot)$, and $\mathcal{L}_{\text{EC-CM}}$ be defined by the Binary Cross Entropy, $BCE(\cdot, \cdot)$. Let $\mathbf{x}_0$ denote the ground truth codeword.*

*First, by the Triangle Inequality of the Total Variation (TV) distance, we have:*

$$\mathcal{L}_{\text{Standard-CM}} = TV(\hat{\mathbf{x}}_t, \hat{\mathbf{x}}_r) \leq TV(\hat{\mathbf{x}}_t, \mathbf{x}_0) + TV(\hat{\mathbf{x}}_r, \mathbf{x}_0). \tag{19}$$

*Squaring both sides of the inequality:*

$$TV^2(\hat{\mathbf{x}}_t, \hat{\mathbf{x}}_r) \leq (TV(\hat{\mathbf{x}}_t, \mathbf{x}_0) + TV(\hat{\mathbf{x}}_r, \mathbf{x}_0))^2$$
$$\leq 2TV^2(\hat{\mathbf{x}}_t, \mathbf{x}_0) + 2TV^2(\hat{\mathbf{x}}_r, \mathbf{x}_0), \tag{20}$$

*By Pinsker's Inequality, which states that for any two probability distributions $P$ and $Q$, $D_{KL}(P\|Q) \geq 2TV^2(P,Q)$,*

$$TV^2(\hat{\mathbf{x}}_t, \hat{\mathbf{x}}_r) \leq D_{KL}(\mathbf{x}_0\|\hat{\mathbf{x}}_t) + D_{KL}(\mathbf{x}_0\|\hat{\mathbf{x}}_r), \tag{21}$$

*Recall that the BCE loss is defined as $BCE(P,Q) = H(Q) + D_{KL}(Q\|P)$, where $H(Q)$ is the entropy of Q. Since entropy is non-negative ($H(Q) \geq 0$), we have:*

$$D_{KL}(\mathbf{x}_0\|\hat{\mathbf{x}}) \leq BCE(\hat{\mathbf{x}}, \mathbf{x}_0), \tag{22}$$

*Substituting them into Eq. 20, we show that:*

$$TV^2(\hat{\mathbf{x}}_t, \hat{\mathbf{x}}_r) \leq BCE(\hat{\mathbf{x}}_t, \mathbf{x}_0) + BCE(\hat{\mathbf{x}}_r, \mathbf{x}_0) = \mathcal{L}_{\textit{EC-CM}}. \tag{23}$$

## A.4 DETAILED EXPERIMENTAL SETTING

We provide the specific design choices of ECCFM and the listed training hyperparameters. Following the design in (Choukroun & Wolf, 2022b; 2023; Park et al., 2024), we generate the training set by corrupting the all-zero codeword $\mathbf{x}_0$ with AWGN noise $z$, following the diffusion process $y = \mathbf{x}_0 + \sqrt{\beta_N} \cdot \epsilon$, where $\epsilon \sim \mathcal{N}(0, I)$. The diffusion process was configured with a total of $N = n - k + 5$ steps. We employed a linear variance schedule with $\beta_i = 0.01$ for short codes and a more fine-grained $\beta_i = 0.0025$ for medium-to-long codes. Each epoch consisted of 1,000 steps with a minibatch size of 128. To ensure a fair comparison, all neural models were implemented with a fixed architecture ($N = 6$ layers, $d = 128$ hidden dimensions). Each test task was run until at least 500 error codes were observed, under a maximum of $10^7$ test instances.

Table 3: Training hyperparameters and design choices.

| Parameters | Design Choice |
|---|---|
| Consistency Loss | $d(\cdot, \cdot) = $ Binary Cross Entropy$(\hat{\mathbf{x}}_0, \mathbf{x}_0)$ |
| Syndrome Weight | $\lambda = 0.01$ |
| Training Epoch | 1500 |
| Mini-batch | 1000 |
| Training Batchsize | 128 |
| Test Numbers | At least 500 error cases and at most $10^7$ total numbers |
| Test Batchsize | 2048 for short codes, 256 for medium-to-long codes |
| Weighting Function | $w(t) = 1$ |
| Total Diffusion Steps | $N = n - k + 5$ |
| Forward Schedule | $\beta_i = 0.01$ for short codes, $\beta_i = 0.0025$ for medium-to-long codes |
| Time Step | $t \sim \mathcal{U}\{1, 2, \ldots, N\}$ |
| Scaling Factor | $\alpha = 0.8$ |
| Initial Learning Rate | $\eta = 1e^{-4}$ |
| Learning Rate Schedule | Cosine Decay |
| Decay Rate | $\eta' = 5e^{-7}$ |
| Exponential Moving Average Ratio | EMA$= 0.999$ |

## A.5 ADDITIONAL EXPERIMENTAL RESULTS

### A.5.1 ADDITIONAL RESULTS ON MULTI-STEP SAMPLING

We investigated the multi-step sampling technique from (Song et al., 2023) to explore the quality-cost trade-off. Using a 2-step strategy with a re-noising ratio of $\alpha = 0.2N$, we observed that performance slightly deteriorated (higher BER) compared to one-step decoding in Table 4.

We interpret this result from the nature of ECC task. Unlike image generation, error correction aims to recover the exact original data from different noise levels, and the multiple decoding steps for a one-step consistency decoder risk introducing additional error rather than refining the result, potentially leading to error accumulation.

Table 4: Performance comparison of ECCFM(1-step) and ECCFM(2-step)

| Code Type | ECCFM (1-step) | | | ECCFM (2-step) | | |
|---|---|---|---|---|---|---|
| $E_b/N_0$ | 4 | 5 | 6 | 4 | 5 | 6 |
| POLAR(64,32) | **7.55** | **10.31** | **13.80** | 7.21 | 9.97 | 13.22 |
| POLAR(128,64) | **8.01** | **12.22** | **16.71** | 7.67 | 11.89 | 16.02 |
| POLAR(128,96) | **7.21** | **10.52** | **14.32** | 6.97 | 10.16 | 13.88 |
| LDPC(121,70) | **7.35** | **12.23** | **17.60** | 7.02 | 11.17 | 15.93 |
| LDPC(121,80) | **8.25** | **13.33** | **18.69** | 7.95 | 13.14 | 17.78 |

### A.5.2 SOFT-SYNDROME UNDER LOW SNR

The soft-syndrome time condition in the diffusion denoising phase does not require any theoretical assumption on SNRs. To validate the efficacy of soft-syndrome condition under low SNR, we visualize the syndrome change for both hard-syndrome and soft-syndrome for POLAR(64,48) and BCH(63,36) codes under 2 dB in Figure 7. The results show that soft-syndrome maintains consistent performance in building a smooth trajectory.

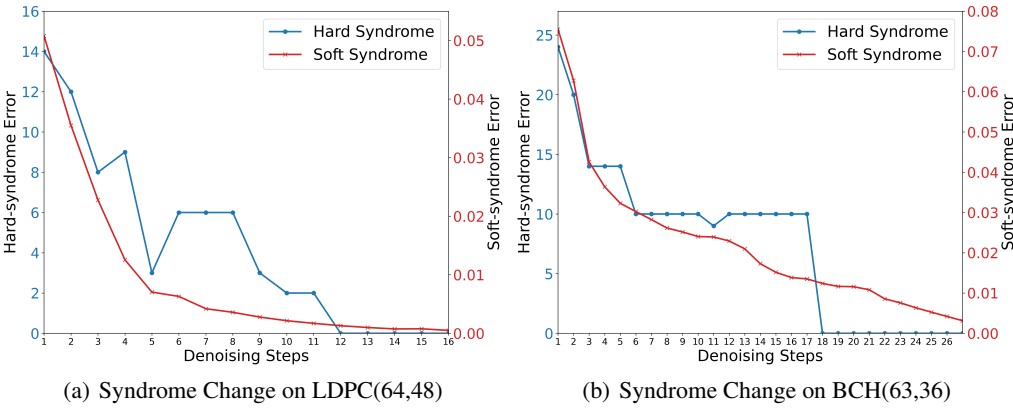

(a) Syndrome Change on LDPC(64,48)   (b) Syndrome Change on BCH(63,36)

Figure 7: Syndrome Change (Hard syndrome and soft syndrome) during iterative denoising on LDPC and BCH under low SNRs (2 dB).

### A.5.3 QUANTIFYING THE TRAINING COSTS

In the main part, we apply the cross-attention transformer proposed by Park et al. (2024) as the backbone architecture for ECCFM. The total number of parameters marginally exceeds that of the standard CrossMPT backbone due to a 2-layer time embedding for the soft-syndrome time condition. We present a comparison of the training costs in Table 5.

### A.5.4 ABLATION STUDY: SOFT-SYNDROME LOSS TERM IN TOTAL LOSS

The soft-syndrome loss term in Eq. 10 serves primarily as a regularization term, designed to stabilize training and accelerate convergence. By explicitly incorporating this loss, we guide the optimization toward a feasible solution space during the early stages of training. To empirically validate this, we conducted an ablation study comparing the model's performance with and without the soft-syndrome term. As shown in Table 6, including this loss results in faster convergence of the primary consistency objective.

Table 5: Quantifying the training costs of ECCFM and CrossMPT.

| Code Type | Parameter | FLOPs | | Model Parameters | | Training (epoch) | |
|-----------|-----------|-------|---|------------------|---|------------------|---|
| | | CrossMPT | ECCFM | CrossMPT | ECCFM | CrossMPT | ECCFM |
| BCH | (61,36) | 112.2 M | 112.6 M | 1.20 M | 1.42 M | 41 s | 71 s |
| POLAR | (64,32) | 120.4 M | 120.8 M | 1.20 M | 1.42 M | 41 s | 75 s |
| | (128,64) | 253.4 M | 253.8 M | 1.24 M | 1.45 M | 43 s | 78 s |
| | (128,96) | 202.8 M | 203.2 M | 1.23 M | 1.44 M | 47 s | 82 s |
| LDPC | (121,70) | 229.6 M | 230.0 M | 1.23 M | 1.44 M | 43 s | 67 s |
| | (121,80) | 212.4 M | 212.9 M | 1.23 M | 1.44 M | 45 s | 69 s |
| WRAN | (384,320) | 608.2 M | 608.6 M | 1.42 M | 1.63 M | 74 s | 83 s |

Table 6: Comparison of training time to achieve the target consistency loss with or without soft-syndrome loss term

| Code Type | Loss function | Target consistency loss | Epoch |
|-----------|---------------|-------------------------|-------|
| POLAR(64,32) | **w/ Soft-syn Loss** | $1.05 \times 10^{-4}$ | **721** |
| | w/o Soft-syn Loss | $1.05 \times 10^{-4}$ | 1323 |
| POLAR(128,64) | **w/ Soft-syn Loss** | $2.06 \times 10^{-2}$ | **866** |
| | w/o Soft-syn Loss | $2.06 \times 10^{-2}$ | 1474 |
| LDPC(121,80) | **w/ Soft-syn Loss** | $6.50 \times 10^{-3}$ | **827** |
| | w/o Soft-syn Loss | $6.50 \times 10^{-3}$ | 1451 |

### A.5.5 ADDITIONAL RESULTS OF SOFT-SYNDROME TIME CONDITION IN CONSISTENCY TRAINING

As stated in the main part, we trained ECCFM with either a hard-syndrome time condition or a soft-syndrome time condition and validated the necessity of using soft-syndrome as time condition for successful training of ECCFM in Table 7.

Table 7: Performance comparison of ECCFM(Hard-syndrome) and ECCFM(Soft-syndrome)

| Code Type | Hard-syndrome | | | Soft-syndrome | | |
|-----------|---------------|---|---|---------------|---|---|
| $E_b/N_0$ | 4 | 5 | 6 | 4 | 5 | 6 |
| POLAR(64,32) | 4.36 | 5.78 | 9.81 | **7.55** | **10.31** | **13.80** |
| POLAR(128,64) | 4.87 | 7.92 | 10.10 | **8.01** | **12.22** | **16.71** |
| POLAR(128,96) | 4.11 | 6.97 | 9.22 | **7.21** | **10.52** | **14.32** |
| LDPC(121,70) | 4.32 | 8.31 | 11.41 | **7.35** | **12.23** | **17.60** |
| LDPC(121,80) | 5.09 | 8.95 | 11.90 | **8.25** | **13.33** | **18.69** |

### A.5.6 APPLYING VANILLA CONSISTENCY TRAINING IN ECC

To clarify the contribution of our proposed consistency training framework, we conducted an ablation study comparing ECCFM against a baseline utilizing the standard consistency model (Vanilla-CM) objective. In this experiment, we retained the forward sampling and time-conditioning mechanisms from DDECC but replaced the training objective with the standard consistency loss following (Song et al., 2023). As shown in Table 8, directly applying the Vanilla-CM to the ECC domain results in a significant performance degradation, and highlights a fundamental mismatch between the

standard consistency objective and the discrete, non-differentiable nature of the syndrome condition in ECC domain.

Table 8: Performance comparison of Vanilla-CM and ECCFM

| Code Type $E_b/N_0$ | Vanilla-CM | | | ECCFM | | |
|---|---|---|---|---|---|---|
| | **4** | **5** | **6** | **4** | **5** | **6** |
| POLAR(64,32) | 4.67 | 6.21 | 8.32 | **7.55** | **10.31** | **13.80** |
| POLAR(128,64) | 4.98 | 6.45 | 8.78 | **8.01** | **12.22** | **16.71** |
| POLAR(128,96) | 4.33 | 5.86 | 7.91 | **7.21** | **10.52** | **14.32** |
| LDPC(121,70) | 4.21 | 6.13 | 8.45 | **7.35** | **12.23** | **17.60** |
| LDPC(121,80) | 4.87 | 6.84 | 9.11 | **8.25** | **13.33** | **18.69** |

### A.5.7 ADDITIONAL RESULTS ON LONG CODES

As established in Figure 4, ECCFM demonstrates scalability, achieving competitive performance on both short and medium-to-long codes. To further validate this, we present additional results for LDPC codes of varying lengths and rates, specifically LDPC($n = 204, k = 102$) and LDPC($n = 408, k = 204$). The BER and FER results in Figure 8 confirm that ECCFM improves decoding performance while maintaining its high inference speed.

### A.5.8 ADDITIONAL RESULTS ON INFERENCE TIME AND THROUGHPUT

To evaluate the practical efficiency of ECCFM, we benchmarked its inference time and throughput on both POLAR and LDPC codes against established baselines (ECCT, CrossMPT, and DDECC). Inference time was measured as the total duration in seconds to decode $10^6$ samples, while throughput was defined as the number of samples decoded per second. As illustrated in Figure 5, ECCFM achieves a speedup over the denoising diffusion method, DDECC. The advantage scales with code complexity, growing from a 30x speedup on short codes to over 100x on longer codes. Further analysis, detailed in Figure 9, Figure 10, Figure 11, and Figure 12 confirms that these efficiency gains are consistent across a wide range of code types, lengths, and rates. This performance demonstrates that ECCFM provides a significant improvement in decoding speed, particularly for long codes where latency is a critical bottleneck.

### A.5.9 ANALYSIS ON COMPUTATIONAL OVERHEAD OF ITERATIVE DENOISING PHASE

To point out the ECCFM's speed advantage, we analyzed the iterative convergence of the denoising diffusion framework. Specifically, we measured the average number of inference steps required for the DDECC model to converge to a valid codeword (i.e., achieve a zero syndrome, $e_t = 0$). As detailed in Table 9, the computational overhead for DDECC increases substantially under these three conditions: longer codes, lower code rates, and lower SNRs. This is precisely the bottleneck that ECCFM's one-step decoding offers a consistent gain in efficiency, particularly in these difficult decoding scenarios.

## A.6 STATEMENT OF LLM USAGE

Large Language Models (LLMs) were used to aid in the proofreading and polishing of the writing of the manuscript.

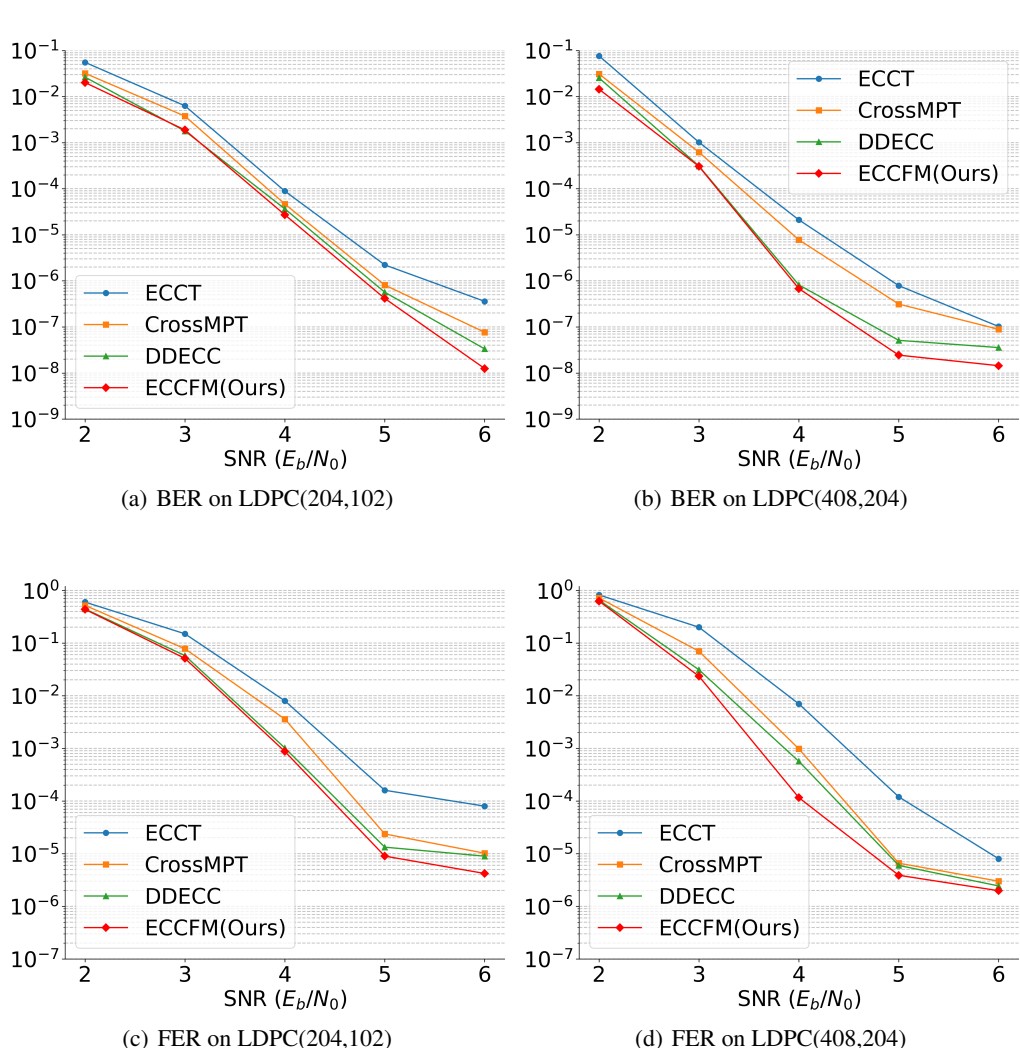

Figure 8: Performance comparison in terms of Bit Error Rate (BER) and Frame Error Rate (FER) for two LDPC codes with different blocklengths and rates: LDPC($n = 204, k = 102$) and LDPC($n = 408, k = 204$). Our method is evaluated against ECCT, CrossMPT, and DDECC.

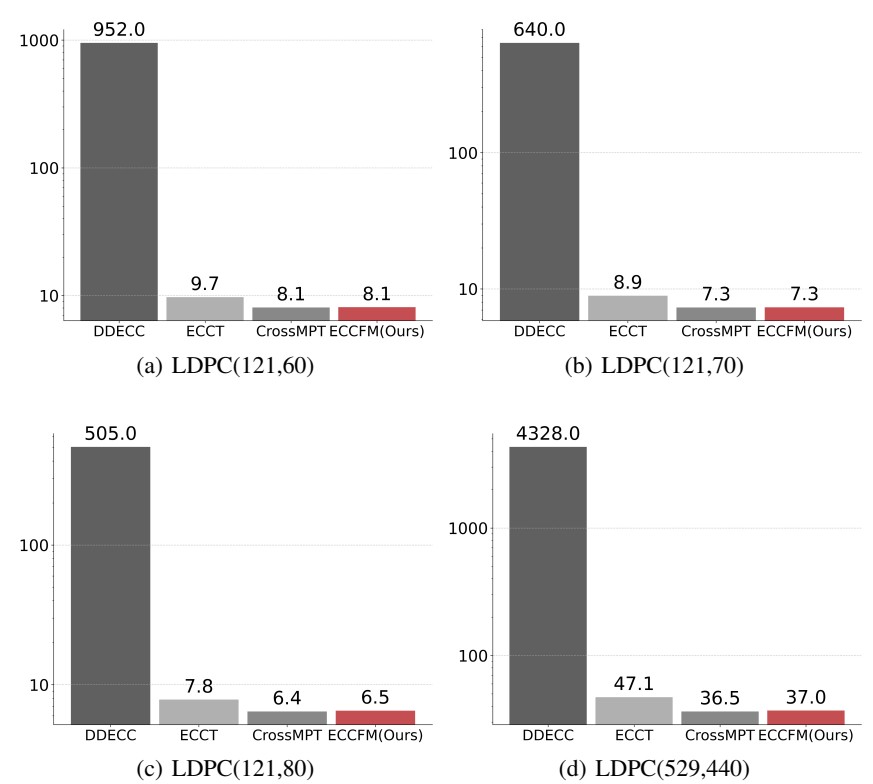

Figure 9: Inference time on LDPC($n = 121, k = 60$), LDPC($n = 121, k = 70$), LDPC($n = 121, k = 80$) and LDPC($n = 529, k = 440$), comparing with ECCT, CrossMPT and DDECC.

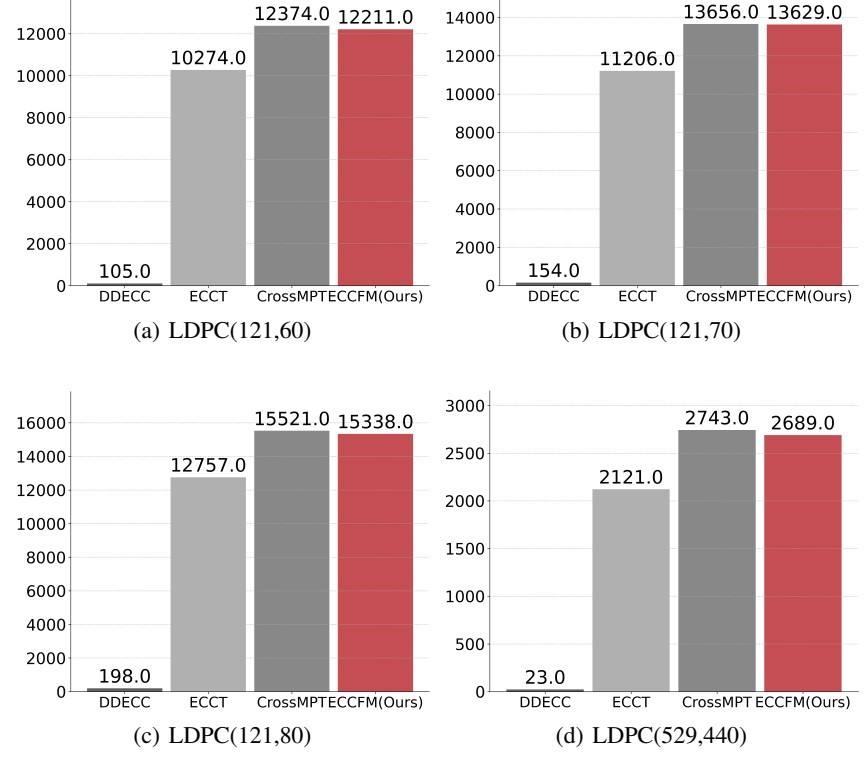

Figure 10: Throughput on LDPC($n = 121, k = 60$), LDPC($n = 121, k = 70$), LDPC($n = 121, k = 80$) and LDPC($n = 529, k = 440$).

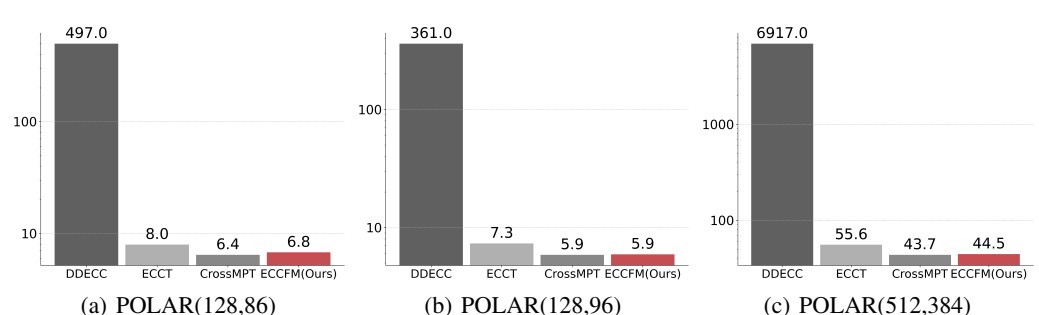

Figure 11: Inference time on POLAR($n = 128, k = 86$), POLAR($n = 128, k = 96$) and POLAR($n = 512, k = 384$), comparing with ECCT, CrossMPT and DDECC.

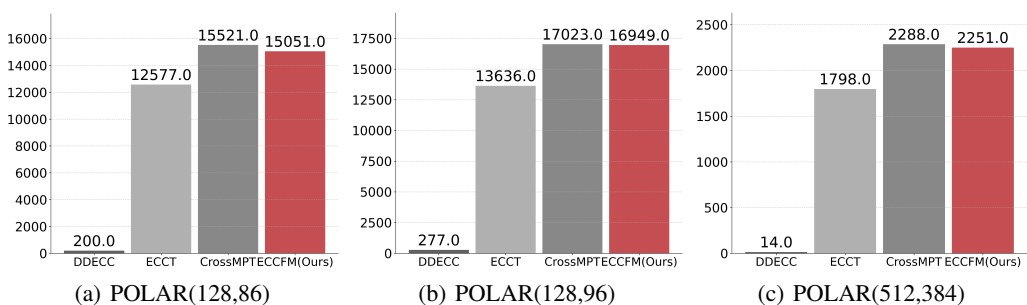

Figure 12: Inference time on POLAR($n = 128, k = 86$), POLAR($n = 128, k = 96$) and POLAR($n = 512, k = 384$).

Table 9: Convergence steps to $e_t = 0$ of the DDECC decoder on longer codes across different Signal-to-Noise Ratios ($E_b/N_0$). The results are reported in terms of the average steps(variance).

| Code Type | Parameters | Converge Steps: Average(Variance) | | | | |
|---|---|---|---|---|---|---|
| | | **2** | **3** | **4** | **5** | **6** |
| **POLAR** | (512,384) | 123.40(11.38) | 91.34(21.68) | 60.24(18.00) | 41.40(16.13) | 24.99(14.82) |
| **LDPC** | (204,102) | 57.10(21.51) | 39.37(10.50) | 29.47(7.50) | 21.25(6.90) | 14.24(6.31) |
| | (408,204) | 112.12(39.47) | 76.91(13.18) | 58.39(10.38) | 42.39(9.62) | 28.99(8.81) |
| | (529,440) | 89.54(8.07) | 59.25(27.75) | 27.48(9.78) | 17.06(6.95) | 9.22(6.03) |
| **WRAN** | (384,320) | 59.66(9.79) | 37.59(18.88) | 18.57(7.99) | 11.23(5.49) | 5.91(4.64) |

