# OpenReview forum: "Consistency Flow Model Achieves One-step Denoising Error Correction Codes"
_ICLR.cc/2026/Conference — Submitted to ICLR 2026_

### Official Review · Reviewer_z34F · 2025-10-29

**Soundness:** 3
**Presentation:** 2
**Contribution:** 2
**Rating:** 2
**Confidence:** 4

**Summary:**

This paper explores the use of diffusion models for the task of error correction code (ECC). While prior works have demonstrated promising results using diffusion models for ECC, they suffer from high computational costs. To improve efficiency, the authors adopt the consistency model, which enables one-step denoising and significantly reduces computational overhead. Experimental results show that the proposed framework achieves a lower bit-error rate compared to previous methods.

**Strengths:**

* The paper is easy to follow.
* The work focuses on an interesting and important application.
* Experimental results indicate that the proposed method achieves promising performance with improved efficiency.

**Weaknesses:**

* Notation
    * The definition of the function f at Line 223 and in Equation (3) appears problematic. The output of f should be a prediction of the codeword rather than a probability distribution. The correct formulation should align with Algorithm 1, e.g., $d(f_\theta(x_r, r), x_0)$ instead of $d(f_\theta(x_r, r), \delta(x - x_0))$.
    * The notation $L_{\text{Consistency}}$ is inconsistent with the previous line at Algorithm 1—it should be written as $L_{\text{EC-CM}}$.
* Claims Without Sufficient Support
    * An ablation study on the use of soft syndrome is missing. Since this is a key contribution of the paper, a comparison between using soft and hard syndromes should be included to substantiate the claimed advantage. Additionally, it would be valuable to report results when incorporating an explicit soft-syndrome loss in the overall objective to validate the design choice.
    * The paper claims that the proposed objective in Equation (3) provides a stronger learning signal than the conventional objective in Equation (2). However, no experimental evidence is provided to support this claim. It remains unclear whether the proposed objective improves performance, convergence speed, or both.
* Limited Insight
    * While I typically avoid judging contributions solely based on novelty, this paper seems to offer limited conceptual insight. The motivation—to enhance the efficiency of diffusion-based denoisers for ECC—is clear, but the proposed solution (adopting the consistency model) is rather straightforward, as the efficiency benefit of consistency models is already well established. Beyond combining existing techniques such as the consistency model and soft-syndrome mechanism to achieve better empirical results, the paper provides limited generalizable insights for broader applications.
* Inaccurate statement
    * The description from Lines 211–214 appears inaccurate. The Boundary Condition and Self-Consistency are not “naturally inherent” properties of ECC data. Instead, they are intrinsic properties of the consistency model framework, imposed on the function $f_\theta$ through its parameterization and training objective. These properties are data-agnostic, whereas the text incorrectly attributes them to ECC itself.

**Questions:**

* During inference, how is the value of $\sigma$ in Equation (7) determined?

---

> ### Author Response · Authors · 2025-11-25
>
> We sincerely thank Reviewer z34F for the thoughtful feedback on our work. We are glad that the reviewer found our work “easy-to-follow” and to focus on “an interesting and important topic”. Below is our response to the reviewer’s comments and questions.
>
> **1- Ablation study on soft-syndrome**
> To validate the efficacy of the soft-syndrome, we conducted a series of targeted ablation studies. First, we evaluated its impact in the total loss as the regularization term. We trained the ECCFM with and without the soft-syndrome loss, measuring the number of epochs required to reach a target consistency loss. As detailed in the table below, incorporating the soft-syndrome loss accelerates convergence, reducing the required training epochs by nearly half across all tested Polar and LDPC codes. This demonstrates its effectiveness in guiding the model toward a valid solution space more efficiently.
> | Code Type | Loss function | Target consistency loss | Epoch |
> | :--- | :--- | :---: | :---: |
> |**POLAR(64,32)** | **w/ soft-syn loss** | $1.05 \times 10^{-4}$ | **721** |
> | **POLAR(64,32)** | w/o soft-syn loss | $1.05 \times 10^{-4}$ | 1323 |
> | **POLAR(128,64)** | **w/ soft-syn loss** | $2.06 \times 10^{-2}$ | **866** |
> | **POLAR(128,64)** | w/o soft-syn loss | $2.06 \times 10^{-2}$ | 1474 |
> | **LDPC(121,80)** | **w/ soft-syn loss** | $6.50 \times 10^{-3}$ | **827** |
> | **LDPC(121,80)** | w/o soft-syn loss | $6.50 \times 10^{-3}$ | 1451 |
>
> Second, we investigated soft-syndrome as a time condition for building the smooth trajectory. We trained ECCFM with hard-syndrome time condition or soft-syndrome time condition. Our experiments revealed that using soft-syndrome as time condition is necessary for successful training of ECCFM. The continuous nature of the soft-syndrome provides a smooth reverse trajectory to learn, whereas models conditioned on the hard syndrome failed to converge.
>
> | Code Type | Hard-syn ($E_b/N_0=4$)| Hard-syn ($E_b/N_0=5$) | Hard-syn ($E_b/N_0=6$) | Soft-syn ($E_b/N_0=4$) |Soft-syn ($E_b/N_0=5$) | Soft-syn ($E_b/N_0=6$) |
> | :--- | :---: | :---: | :---: | :---: | :---: | :---: |
> | POLAR(64,32) | 4.36 | 5.78 | 9.81 | **7.55** | **10.31** | **13.80** |
> | POLAR(128,64) | 4.87 | 7.92 | 10.10 | **8.01** | **12.22** | **16.71** |
> | POLAR(128,96) | 4.11 | 6.97 | 9.22 | **7.21** | **10.52** | **14.32** |
> | LDPC(121,70) | 4.32 | 8.31 | 11.41 | **7.35** | **12.23** | **17.60** |
> | LDPC(121,80) | 5.09 | 8.95 | 11.90 | **8.25** | **13.33** | **18.69** |

---

> ### Author Response · Authors · 2025-11-25
>
> **2- Insights and Results in our study**
> We appreciate the reviewer’s point that consistency models are well known for improving inference efficiency. In this response, we would like to clarify why bringing the consistency framework to the ECC setting is still an essential and non-trivial task. Following this response, we will update the introduction to better reflect the significance and non-triviality of the task and method.
> First, we highlight that the best AI decoder performance in the literature is currently achieved by DDECC [1] that is a diffusion-based decoder. However, a limitation of DDECC is the long latency caused by iterative reverse denoising, which is significantly higher than that of non-diffusion AI decoders. In typical wireless communication settings on mobile devices, this latency and the associated computation cost make DDECC’s deployment difficult. Reducing the number of diffusion iterations is therefore necessary if diffusion-based decoders are to be deployed in typical wireless communication settings.
> Continuing our response, we highlight that extending consistency models to this setting remains non-trivial, and a direct application of vanilla consistency models performs significantly worse than DDECC. To demonstrate this, we have conducted numerical experiments comparing Vanilla CM loss training and our proposed EC-CM loss training. The results show that directly applying vanilla consistency models leads to a significant performance drop.
>
> | Code Type | EC-CM ($E_b/N_0=4$) | EC-CM ($E_b/N_0=5$) | EC-CM ($E_b/N_0=6$) | Vanilla-CM ($E_b/N_0=4$) | Vanilla-CM ($E_b/N_0=5$) | Vanilla-CM ($E_b/N_0=6$) |
> | :--- | :---: | :---: | :---: | :---: | :---: | :---: |
> | **POLAR(64,32)** | **7.55** | **10.31** | **13.80** | 4.67 | 6.21 | 8.32 |
> | **POLAR(128,64)** | **8.01** | **12.22** | **16.71** | 4.98 | 6.45 | 8.78 |
> | **POLAR(128,96)** | **7.21** | **10.52** | **14.32** | 4.33 | 5.86 | 7.91 |
> | **LDPC(121,70)** | **7.35** | **12.23** | **17.60** | 4.21 | 6.13 | 8.45 |
> | **LDPC(121,80)** | **8.25** | **13.33** | **18.69** | 4.87 | 6.84 | 9.11 |
>
> Therefore, our main contribution is to show that a carefully designed consistency model can recover the performance of DDECC with much lower inference time. We achieve this by introducing a training objective tailored to ECC decoding, by proposing soft syndrome as a continuous-time condition to ensure a smooth trajectory during training, and by incorporating soft syndrome regularization to stabilize and speed up convergence. Without these elements, consistency models remain considerably below the performance of DDECC. With the elements included, we could match its accuracy at significantly lower latency, making diffusion-based decoding more feasible for devices with strict computation and delay limits.
>
> **3- Statement of consistency properties**
> We would like to clarify that our statement in Lines 208-215 aims to highlight what the boundary and self-consistency conditions mean in the ECC context and why we expect them to hold naturally. We agree with the reviewer that the properties can similarly hold in other domains, and will make this point clear that the statement is to explain how to interpret the conditions in the context of ECC.
>
> **4- Notations**
> We thank Reviewer z34F for pointing out the inconsistency in notations. We will carefully correct the notations in the revised manuscript.
>
> **5- Value of $\sigma$**
> The noise level $\sigma$ is given to the decoder at the receiver's side, as an input of the algorithm. In practice under the 5G standard, reference signals will be sent to estimate the quality of the communication channels (3GPP TS38.211 Clause 6.4.1) and report the channel state information (3GPP TS38.211 Clause 5.2).

---

> > ### Comment · Reviewer_z34F · 2025-11-26
> >
> > The authors’ response addresses some of my concerns, and I have adjusted my rating accordingly. However, several important issues remain unresolved.
> >
> > First, during the rebuttal period, authors are allowed to upload an updated version. It would be very helpful to see the promised revisions reflected in an updated manuscript.
> >
> > I remain concerned about the claimed advantage of Eq. 3 over Eq. 2 from the original LCM paper. In fact, Eq. 3 appears to be equivalent to the standard diffusion loss, and there seems to be no need to sample different r and t when all time steps are supervised to predict x_0. Additionally, the reported performance of Vanilla-CM is surprisingly low—substantially worse than the diffusion baseline. This raises concerns about an unfair comparison. More details and justification are needed to explain this large performance gap, especially given that LCM does not underperform standard diffusion models in image generation.
> >
> > Similarly, the ablation on soft syndrome shows a very large improvement, yet the hard-syndrome variant performs unexpectedly poorly. Since DDECC also uses hard syndrome and achieves much stronger results, this discrepancy raises concerns about the implementation and requires further clarification.
> >
> > Finally, I would like to reiterate that the consistency properties do not depend on any specific data type. They arise solely from the parameterization and training objective introduced in LCM. The manuscript’s current phrasing is therefore misleading and should be corrected.

---

> > > ### Author Response · Authors · 2025-11-29
> > >
> > > **1- Revised manuscript**
> > >
> > > We have uploaded a revised manuscript incorporating the changes we discussed in our responses to this and the other reviews.
> > >
> > > **2- On the Role of Soft-Syndrome in Enhancing Consistency Model ECC Performance**
> > >
> > > We would like to clarify that our numerical results consistently show that introducing soft-syndrome as the time conditioning provides a substantial performance gain and enables Consistency Models to operate effectively in the ECC setting. This improvement is well-motivated by the underlying mechanics of Consistency Models: these models learn a mapping along a Probability-Flow ODE (PF-ODE), a process that fundamentally relies on smooth and differentiable trajectories in time.
> > > In standard reverse diffusion for image generation, the denoising trajectory moves toward a continuous image data manifold. However, in ECC decoding, the target structure is **discrete**, and a direct hard-syndrome conditioning forces the model into **non-smooth transitions**. This mismatch makes it difficult for a PF-ODE–based model to learn an accurate and stable flow. By replacing hard-syndrome with a soft, differentiable alternative, we could restore the smoothness required by the PF-ODE formulation, which explains the strong empirical improvement observed in our experiments.
> > >
> > > This distinction can also explain the numerical observation regarding DDECC. In the DDECC framework, the hard syndrome serves as a time condition label indicating the noise level, and the discrete nature of the hard syndrome does not significantly hinder its training. In contrast, the consistency model must map any $x_t$ to $x_0$ in a single step, which strictly requires the smooth trajectory provided by our soft syndrome. Therefore, the performance drop in our hard syndrome ablation is expected and validates the necessity of our contribution.
> > >
> > > We hope this explanation clarifies that the performance gaps are driven by the discrete nature in ECC domains and highlights the necessity of soft syndrome as one of the core innovations of our work.

---

### Official Review · Reviewer_F3Pq · 2025-10-30

**Soundness:** 3
**Presentation:** 2
**Contribution:** 2
**Rating:** 4
**Confidence:** 4

**Summary:**

The authors address the long-latency problem of DDECC by introducing a consistency flow model. In addition, they employ a soft-syndrome formulation to replace the hard-syndrome approach. The results are meaningful, achieving better performance than the baseline, CrossMPT, while maintaining similar inference time.

**Strengths:**

The motivation of the paper is clear, and the methodology is well adapted from the machine learning field. The modification of the model for the channel coding context is also well derived. The simulation results overall appear to be accurate and convincing.

**Weaknesses:**

Most parts of the paper focus on the training method. However, I am curious about the decoding architecture. The authors mention that CrossMPT is used — does this mean the architecture is exactly the same as CrossMPT? How does e_T (the second parameter of f_theta) affect the decoding process? Do we need multiple models depending on the value of this second parameter to perform decoding? If so, this could be a drawback of the proposed approach. Please clarify this point.

In the main body of the paper, it would be valuable to include simulation results for ECCFM with the ECCT architecture, not only with the CrossMPT architecture. For a fair comparison with DDECC, both ECCFM and DDECC should share the same underlying architecture (ECCT). Although I noticed that the authors included results with the ECCT-based architecture in the Appendix, it would be better to include them in the main text.

The graphs in Figure 4 (particularly the FER graph for Polar code) appear somewhat abnormal. Increasing the SNR step size resolution from 1 dB to 0.5 dB and raising the maximum number of testing trials from 10^7 to 10^8 would improve the reliability of the results.

**Questions:**

The sentence above Eq. (6) mentions the “soft-syndrome error condition for each row j,” but the formulation of the soft-syndrome error condition sums over all j, losing the dependency on j. Could the authors clarify this inconsistency?

In Eq. (8), what is the specific reason for including the soft-syndrome loss in the total loss term? Is it primarily for stabilizing training, or does it also contribute to performance improvement?

Many com fair comparison, the number of training epochs should be aligned.parative works use 1000 epochs, while this paper uses 1500. For a

---

> ### Author Response · Authors · 2025-11-25
>
> We sincerely thank Reviewer F3Pq for the thoughtful feedback and suggestions on our work. We are glad that the reviewer found the motivation of our work “clear” and the simulation results to be “accurate and convincing”. Below is our response to the reviewer’s comments and questions
>
> **1- Decoding architecture**
> We thank Reviewer F3Pq for this point, and we will clarify the decoding architecture in the paper. Our approach uses a single backbone model for all decoding. The time condition $e_T$ is the soft-syndrome, which is directly calculated from the received signal at inference time and does not require separate models. In the main text, ECCFM uses CrossMPT or ECCT as a backbone, while feeding the computed soft-syndrome $e_T$ into a 2-layer MLP as the time embedding. This time embedding adds minimal parametric overhead compared to the original backbone.
>
> **2- Clarity of figures’ presentation**
> Following the reviewer’s comment, in the revision, we will revise the figure’s presentation and increase the upper limit of testing codes to $10^8$ and reduce the SNR interval length from 1 dB to 0.5dB.
>
> **3- Formulation in Eq.(6)**
> We would like to emphasize the usage of $s_j^\dagger$, which is the probability of syndrome at row j of a received codeword being zero. Under the mean field assumption, we can assume that the probability of syndrome for different row j is independent, thus we use a BCE loss to aggregate the syndrome loss for each row together.
>
> **4- Soft-syndrome loss term**
> We note that the soft-syndrome loss term is primarily a regularization term designed to stabilize and accelerate the convergence during the training process. By explicitly training the model with soft-syndrome loss, we guide the optimization towards a feasible solution space early in training.
> To validate this empirically, we have conducted an ablation study of the presence and absence of the soft-syndrome loss term. The results in the following table demonstrate that including the soft-syndrome loss results in faster convergence of the primary consistency objective.
> | Code Type | Loss function | Target consistency loss | Epoch |
> | :--- | :--- | :---: | :---: |
> |**POLAR(64,32)** | **w/ soft-syn loss** | $1.05 \times 10^{-4}$ | **721** |
> | **POLAR(64,32)** | w/o soft-syn loss | $1.05 \times 10^{-4}$ | 1323 |
> | **POLAR(128,64)** | **w/ soft-syn loss** | $2.06 \times 10^{-2}$ | **866** |
> | **POLAR(128,64)** | w/o soft-syn loss | $2.06 \times 10^{-2}$ | 1474 |
> | **LDPC(121,80)** | **w/ soft-syn loss** | $6.50 \times 10^{-3}$ | **827** |
> | **LDPC(121,80)** | w/o soft-syn loss | $6.50 \times 10^{-3}$ | 1451 |
>
> **5- Number of training epochs**
> We would like to clarify that in our numerical analysis, we have already implemented all baseline models and trained them under the optimal setting (1000 epochs for ECCT, CrossMPT, and 2000 epochs for DDECC), in order to ensure a fair comparison. The duration was chosen, because it was sufficient to ensure that every method has reached full convergence. We also numerically report the comparison of training costs between ECCFM and the most efficient CrossMPT:
> | Code Type | FLOPs (CrossMPT) | **FLOPs (ECCFM)** | Params (CrossMPT) | **Params (ECCFM)** | Training per Epoch (CrossMPT) | **Training per Epoch (ECCFM)** |
> | :--- | :---: | :---: | :---: | :---: | :---: | :---: |
> | **BCH(61,36)** | 112.2 M | 112.6 M | 1.20 M | 1.42 M | 41 s | 71 s |
> | **POLAR(64,32)** | 120.4 M | 120.8 M | 1.20 M | 1.42 M | 41 s | 75 s |
> | **POLAR(128,64)** | 253.4 M | 253.8 M | 1.24 M | 1.45 M | 43 s | 78 s |
> | **POLAR(128,96)** | 202.8 M | 203.2 M | 1.23 M | 1.44 M | 47 s | 82 s |
> | **LDPC(121,70)** | 229.6 M | 230.0 M | 1.23 M | 1.44 M | 43 s | 67 s |
> | **LDPC(121,80)**| 212.4 M | 212.9 M | 1.23 M | 1.44 M | 45 s | 69 s |
> | **WRAN(384,320)** | 608.2 M | 608.6 M | 1.42 M | 1.63 M | 74 s | 83 s |

---

### Official Review · Reviewer_LFFA · 2025-11-01

**Soundness:** 3
**Presentation:** 3
**Contribution:** 3
**Rating:** 6
**Confidence:** 2

**Summary:**

This paper introduced a new architecture-agnostic training framework for high-fidelity one-step decoding. It seems that this work integrates the consistency model framework to transformer-based decoders well.

**Strengths:**

- This paper is the first to apply the consistency model framework to error-correcting codes (ECC), achieving state-of-the-art performance.
- The proposed approach replaces the reverse process of the diffusion model with consistency model framework, effectively improving inference efficiency and reducing overall latency compared to DDECC.
- For the noise condition, this paper employs soft-syndromes, whereas DDECC uses hard syndromes, resulting in smoother trajectories and more stable training.

**Weaknesses:**

- In the modifying the loss function, the authors applied triangle inequality. However, since binary cross entropy (BCE) is not a distance metric, it is unclear whether applying the triangle inequality is theoretically valid in this context.
- As the experiments were conducted only on the ECCT architecture, it may be inappropriate to claim the model-agnostic properties. Additional results using other architectures, such as CNN would strengthen this claim.
- When adopting the consistency model (CM) framework, it would be helpful to quantify how much the training cost was reduced compared to other models. A more detailed analysis—such as reporting FLOPs or the number of parameters—would also be beneficial.
- The texts in Figure 5,7,8,9, and 10 are too small and difficult to read.
- Equation (6) should add a minus sign at the front, since it represents the binary cross-entropy between the estimated syndrome and the all-zero syndrome.
- In Equation (7), the “+” following the 1/2 should be a “–”. This is because, under BPSK modulation, a valid codeword satisfies the parity condition with an even number of ones, resulting in a soft syndrome value of 0. If we follow equation (7), the soft syndrome becomes 1 when the codeword is valid, which contradicts the statement in the paper.

**Questions:**

- The paper states that ECCFM employs a Transformer architecture with cross-attention. However, it is unclear which specific model was used. Is this architecture distinct from CrossMPT or a variant of it? If it differs from CrossMPT, please include the performance of the neural decoder used in ECCFM for comparison.
- In Equation (7), the “+” following the 1/2 should be a “–”. Under BPSK modulation, a valid codeword satisfies the parity condition with an even number of ones, leading to a soft syndrome value of 0. According to the current formulation in Equation (7), a valid codeword yields a soft syndrome of 1, which contradicts the intended parity-check behavior.
- In Table I, the results are presented only as numerical values. It would be beneficial to include corresponding figures for representative cases to enhance interpretability and comparison.

---

> ### Author Response · Authors · 2025-11-25
>
> We sincerely thank Reviewer LFFA for the careful and thoughtful feedback and suggestions on our work. We are pleased that the reviewer found our method “effectively improving inference efficiency and reducing overall latency compared to DDECC”. Below is our response to the reviewer’s comments and questions
>
> **1- Triangle inequality in loss function**
> We thank Reviewer LFFA for the feedback. As pointed out by the reviewer, we would like to clarify that the Binary Cross-Entropy (BCE) loss is not a distance metric and thus may not satisfy the triangle inequality. To address this comment, we propose leveraging Pinsker’s Inequality to propose a semi-triangle inequality for the BCE loss where the lower bound is in terms of the total variation distance.
> To clarify the structure behind our objective, we use the fact that for Bernoulli variables, BCE upper-bounds the KL divergence because $BCE(p,x_{0}) = H(x_{0}) + KL(x_{0}|p)$ and the entropy term $H(x_{0})$ is always non-negative. Combining this with Pinsker’s inequality, which relates KL to the total variation (TV) distance, yields the following semi–triangle inequality:
> $$ \mathrm{TV}^{2}\bigl(f\_\theta(\mathbf{x}\_{t},t), f\_\theta(\mathbf{x}\_{r},r)\bigr)
> \le \mathrm{BCE}\bigl(f\_\theta(\mathbf{x}\_{t},t),\mathbf{x}\_{0}\bigr)
> +\mathrm{BCE}\bigl(f\_\theta(\mathbf{x}\_{r},r),\mathbf{x}\_{0}\bigr).$$
> This shows that although BCE itself is not a metric distance, our formulation still controls the valid TV metric, providing an analytical justification for our proposed loss. We also conducted experiments to confirm the validity of this loss design. We compare this training objective to a vanilla consistency loss, our design demonstrates better performance for training a consistency model. This empirical evidence shows that this structurally-inspired loss leads to better performance.
> | Code Type | EC-CM ($E_b/N_0=4$) | EC-CM ($E_b/N_0=5$) | EC-CM ($E_b/N_0=6$) | Vanilla-CM ($E_b/N_0=4$) | Vanilla-CM ($E_b/N_0=5$) | Vanilla-CM ($E_b/N_0=6$) |
> | :--- | :---: | :---: | :---: | :---: | :---: | :---: |
> | **POLAR(64,32)** | **7.55** | **10.31** | **13.80** | 4.67 | 6.21 | 8.32 |
> | **POLAR(128,64)** | **8.01** | **12.22** | **16.71** | 4.98 | 6.45 | 8.78 |
> | **POLAR(128,96)** | **7.21** | **10.52** | **14.32** | 4.33 | 5.86 | 7.91 |
> | **LDPC(121,70)** | **7.35** | **12.23** | **17.60** | 4.21 | 6.13 | 8.45 |
> | **LDPC(121,80)** | **8.25** | **13.33** | **18.69** | 4.87 | 6.84 | 9.11 |
>
> **2- Model architecture**
> We would like to clarify that ECCFM does not depend on the specific model, making it compatible with any underlying architecture. We validated this by testing our method on two different architectures: ECCT and CrossMPT, which are both established methods in the literature. In the main part, we conducted experiments using CrossMPT as the backbone network with a time-embedding layer as the time condition. In Appendix Table 3, we show the results using ECCT as the backbone. In both cases, ECCFM shows consistent performance gains over the baseline transformer-based methods.
>
> **3- Quantifying training costs**
> The number of model parameters is slightly greater than that of the most efficient CrossMPT backbone due to the time-embedding layer for soft-syndrome. We report the comparison of training costs between different methods.
> | Code Type | FLOPs (CrossMPT) | **FLOPs (ECCFM)** | Params (CrossMPT) | **Params (ECCFM)** | Training per Epoch (CrossMPT) | **Training per Epoch (ECCFM)** |
> | :--- | :---: | :---: | :---: | :---: | :---: | :---: |
> | **BCH(61,36)** | 112.2 M | 112.6 M | 1.20 M | 1.42 M | 41 s | 71 s |
> | **POLAR(64,32)** | 120.4 M | 120.8 M | 1.20 M | 1.42 M | 41 s | 75 s |
> | **POLAR(128,64)** | 253.4 M | 253.8 M | 1.24 M | 1.45 M | 43 s | 78 s |
> | **POLAR(128,96)** | 202.8 M | 203.2 M | 1.23 M | 1.44 M | 47 s | 82 s |
> | **LDPC(121,70)** | 229.6 M | 230.0 M | 1.23 M | 1.44 M | 43 s | 67 s |
> | **LDPC(121,80)**| 212.4 M | 212.9 M | 1.23 M | 1.44 M | 45 s | 69 s |
> | **WRAN(384,320)** | 608.2 M | 608.6 M | 1.42 M | 1.63 M | 74 s | 83 s |
>
> **4- Signs in Eq.(6) and Eq.(7)**
> We thank Reviewer LFFA for pointing out the typos, and we will correct the signs in the revised manuscript.
>
> **5- Visualized results**
> We thank Reviewer LFFA for this valuable suggestion. While the results in Table 1 follow the reporting standards of prior works (ECCT, DDECC, CrossMPT), we will include more detailed figures showing the performance in the revised manuscript.

---

### Official Review · Reviewer_teDY · 2025-11-09

**Soundness:** 2
**Presentation:** 3
**Contribution:** 2
**Rating:** 4
**Confidence:** 2

**Summary:**

This paper introduces the Error Correction Consistency Flow Model (ECCFM), a framework that enables single-step decoding for error correction codes. By using an "soft syndrome" condition, it matches the state-of-the-art accuracy of slow diffusion-based decoders while delivering massive 30-100x speedups, making high-performance neural decoding practical for low-latency applications.

**Strengths:**

- The method innovatively introduces a 'soft syndrome' to solve the critical problem of non-smooth trajectories when applying consistency models to ECC decoding.
- It uniquely achieves both state-of-the-art decoding accuracy and massive inference speedups of 30-100x over diffusion-based methods.
- The framework has high practical value for low-latency applications and is model-agnostic, making it widely applicable to different network architectures.
- Its claims are substantiated by rigorous and comprehensive experiments across various standard codes with fair comparisons to strong baselines.

**Weaknesses:**

- The study's evaluation is confined to the ideal AWGN channel, leaving its effectiveness in more realistic fading channels unexplored.
- The training process is potentially sensitive and highly dependent on the construction of a smooth trajectory, which may require careful tuning for different codes.
- Its core idea is a clever adaptation of consistency models from another field, rather than a fundamentally new theoretical invention. I'm more familiar with consistency models than with error-correcting codes, so it's hard to assess the novelty.

**Questions:**

- How robust is the core 'soft syndrome' method under extremely low SNR conditions, where unreliable Log-Likelihood Ratios could disrupt the smoothness of the decoding trajectory and lead to performance degradation?
- Have the authors experimented with multi-step sampling for ECCFM, and if so, does it provide a meaningful improvement in Bit Error Rate, offering a flexible trade-off between decoding latency and accuracy?
- What are the primary technical challenges in extending the ECCFM framework to more complex channel models like Rayleigh fading, and would it require a fundamental redefinition of the 'soft syndrome' to maintain a learnable trajectory?

**Details Of Ethics Concerns:**

None.

---

> ### Author Response · Authors · 2025-11-25
>
> We sincerely thank Reviewer teDY for the thoughtful feedback and suggestions on our work. We are pleased that the reviewer found our work to offer “practical value for low-latency applications” and our claims to be “substantiated by rigorous and comprehensive experiments”. Below is our response to the reviewer’s comments and questions
>
> **1- Training settings**
> We would like to clarify that in our experiments, we did not need to alter the training process significantly by tuning parameters. In our experiments, we kept almost all the hyperparameters to take the same value for different codes. The only parameter we adjusted was the forward noise schedule, i.e. $\beta_i=0.01$ for short codes (length $\leq$ 128) and $\beta_i=0.0025$ for long codes (length > 128). All other key parameters were fixed. Since our method led to consistent performance across all these different codes without considerable code-specific hyperparameter tuning, it shows that the training process does not require significant tuning. We will clarify this point in the revision.
>
> **2- Soft-syndrome under low SNRs**
> We note that the soft-syndrome term does not require any theoretical assumption on SNRs, and we also provide the soft-syndrome curve under POLAR and BCH codes under low SNR (2 dB). The results show that soft-syndrome maintains consistent performance in building a smooth trajectory.
>
> **3- Multi-step Sampling**
> Following the reviewer’s point, we have investigated the multi-step sampling and conducted experiments using a 2-step decoding strategy, which is similar to the multi-step sampling approach proposed in [1]. Our results for different SNRs ($E_b/N_0=\{4,5,6\}$) showed that using a second step slightly decreased the performance, leading to a higher Bit Error Rate (BER) compared to our one-step method.
>
> | Code Type | 1-step ($E_b/N_0=4$) | 1-step ($E_b/N_0=5$) | 1-step ($E_b/N_0=6$) | 2-step ($E_b/N_0=4$) | 2-step ($E_b/N_0=5$) | 2-step ($E_b/N_0=6$) |
> | :--- | :---: | :---: | :---: | :---: | :---: | :---: |
> | **POLAR(64,32)** | **7.55** | **10.31** | **13.80** | 7.21 | 9.97 | 13.22 |
> | **POLAR(128,64)** | **8.01** | **12.22** | **16.71** | 7.67 | 11.89 | 16.02 |
> | **POLAR(128,96)** | **7.21** | **10.52** | **14.32** | 6.97 | 10.16 | 13.88 |
> | **LDPC(121,70)** | **7.35** | **12.23** | **17.60** | 7.02 | 11.17 | 15.93 |
> | **LDPC(121,80)** | **8.25** | **13.33** | **18.69** | 7.95 | 13.14 | 17.78 |
>
> We interpret this result from the nature of ECC task. Unlike image generation, error correction aims to recover the exact original data from different noise levels, and the multiple decoding steps for a one-step consistency decoder risks introducing additional error rather than refining the result, potentially leading to error accumulation.
>
> **4- Complex channels**
> We would like to clarify that addressing the task for more complex channels like Rayleigh fading does not require redefining Soft-syndrome. The construction for a smooth trajectory via Soft-syndrome is also independent of the underlying channel model. We have conducted experiments on Rayleigh channels that indicate competitive results compared to other baselines for different SNRs ($E_b/N_0=\{4,5,6\}$). The numerical results confirm that the existing ECCFM is robust and generalizes effectively without fundamental modification for Rayleigh fading channels.
>
> |Code Type|ECCT|CrossMPT|DDECC|**ECCFM**|
> |---|---|---|---|---|
> |POLAR(64,32)|5.26, 6.27, 7.37|5.87, 7.01, 8.48|5.40, 6.47, 7.64|**5.96, 7.18, 8.67**|
> |POLAR(128,64)|4.24, 5.24, 6.29|4.68, 5.17, 7.30|5.44, 6.90, 8.56|**5.55, 7.22, 8.95**|
> |POLAR(128,96)|3.58, 4.22, 5.02|3.76, 4.52, 5.48|4.21, 5.49, 6.92|**4.75, 5.63, 7.03**|
> |LDPC(121,80)|4.03, 5.06, 6.39|4.31, 5.49, 6.97|4.56, 5.87, 7.50|**4.73, 6.02, 8.01**|
>
> [1] Song Yang, et al. "Consistency Models." ICML 2023.

---

### Author Response · Authors · 2025-12-03
**Wrap-Up: Summary of Clarifications and Revisions**

We sincerely thank the reviewers for their thoughtful and constructive comments, which have helped us further improve the clarity and presentation of the manuscript. Below, we summarize the key points addressed in our responses and revision.

**1- Motivation of our work in reducing latency for diffusion-based ECC on communication devices**

In the response to Reviewer z34F, we clarified that our work addresses a practical challenge of the state-of-the-art DDECC diffusion-based decoder: the high decoding latency caused by iterative denoising steps. For communication systems with strict delay and compute constraints such as mobile devices, lowering inference time is **necessary** for deploying the diffusion-based decoders. Our proposed consistency-based ECCFM framework provides a single-step alternative that substantially reduces latency while maintaining the decoding performance, expanding the feasibility of diffusion-based ECC methods for real world applications.

**2- Importance of our Soft-Syndrome and ECC-Specific design choices in the Consistency Model formulation of DDECC**

In our discussion with Reviewer z34F, we highlighted that Consistency Models require smooth time-conditioned trajectories, which do not hold under the hard-syndrome time conditioning in the DDECC framework. Confirming this intuition, our analysis and ablations also indicate that directly applying a vanilla consistency model leads to a significant performance drop in the ECC setting. We clarified that our proposed **soft-syndrome time condition**  as well as *ECC-specific objective and regularization* collectively provide the smoothness and stability needed for effective consistency training, enabling performance comparable to diffusion-based decoding in DDECC at lower inference cost.

**3- Performance on non-Gaussian channels**

To address the reviewers’ questions, we evaluated our proposed method under Rayleigh fading channels to assess robustness beyond AWGN settings. The results indicate that our proposed ECCFM maintains the performance and compares favorably to established baselines, suggesting that the proposed approach generalizes well across different channel models.

**4- Verification of Model-Agnostic Design**

To examine architecture dependence, we tested ECCFM with multiple backbone decoders, including CrossMPT and ECCT. The method demonstrated consistent improvements across architectures, supporting its applicability as a model-agnostic framework rather than a framework tailored to a specific backbone.

**5- Convergence Behavior and Training Dynamics**

Through controlled ablation studies, we analyzed the effect of the soft-syndrome loss term on training dynamics. The results show faster convergence toward the target consistency objective, indicating that the proposed regularization contributes to more stable and efficient optimization.

In addition, we highlight that our revision incorporates improvements in notation consistency, figure clarity, SNR resolution, and formatting based on the reviewers’ feedback.

---

### Meta-Review · Area_Chair_GHH8 · 2025-12-23

**Summary:**

A reviewer gave score of 6 with lower confidence, other reviewers are inclined to reject the paper because of experiment, insight and notation problem. The paper seems to offer limited conceptual insight, though the paper introduces a soft-syndrome formulation to replace the hard-syndrome approach. The results are meaningful, achieving better performance than the baseline, due to the efficiency benefit of consistency models is already well established.

**Reviewer Concerns:**

The paper seems to offer limited conceptual insight (z34F). The motivation—to enhance the efficiency of diffusion-based denoisers for ECC—is clear, but the proposed solution (adopting the consistency model) is rather straightforward, as the efficiency benefit of consistency models is already well established.

**Reviewer Scores:**

A reviewer gave score of 6 with lower confidence,while other reviewers gave below 4 score.

---

### Decision · Program_Chairs · 2026-01-26

Reject